# PROVABLE ACCURACY BOUNDS FOR
# HYBRID DYNAMICAL OPTIMIZATION AND SAMPLING

**Matthew X. Burns, Qingyuan Hou & Michael C. Huang**
Department of Electrical and Computer Engineering
University of Rochester
Rochester, NY 14527, USA
`{mburns13,qhou3}@ur.rochester.edu`
`michael.huang@rochester.edu`

## ABSTRACT

Analog dynamical accelerators (DXs) are a growing sub-field in computer architecture research, offering order-of-magnitude gains in power efficiency and latency over traditional digital methods in several machine learning, optimization, and sampling tasks. However, limited-capacity accelerators require hybrid analog/digital algorithms to solve real-world problems, commonly using large-neighborhood local search (LNLS) frameworks. Unlike fully digital algorithms, hybrid LNLS has no non-asymptotic convergence guarantees and no principled hyperparameter selection schemes, particularly limiting cross-device training and inference.

In this work, we provide non-asymptotic convergence guarantees for hybrid LNLS by reducing to block Langevin Diffusion (BLD) algorithms. Adapting tools from classical sampling theory, we prove exponential KL-divergence convergence for randomized and cyclic block selection strategies using ideal DXs. With finite device variation, we provide explicit bounds on the 2-Wasserstein bias in terms of step duration, noise strength, and function parameters. Our BLD model provides a key link between established theory and novel computing platforms, and our theoretical results provide a closed-form expression linking device variation, algorithm hyperparameters, and performance.

## 1 INTRODUCTION

Computing research has long borrowed from the physical sciences. Sampling and optimization algorithms such as simulated annealing (Kirkpatrick, 1984), parallel tempering (J. Earl & W. Deem, 2005), and Langevin Monte Carlo (LMC) (Chewi et al., 2021) were directly inspired by physical processes observed in nature. In like spirit, a growing computer architecture sub-field has proposed leveraging physical dynamics to accelerate computationally expensive workloads using "dynamical accelerators" (DXs). Originally, research focused on combinatorial optimization problems (Inagaki et al., 2016; Ushijima-Mwesigwa et al., 2017; Wang & Roychowdhury, 2019; Afoakwa et al., 2021; Mohseni et al., 2022) and matrix-vector multiplication Xiao et al. (2022). However, the field has expanded to sampling for energy-based model training and inference (Vengalam et al., 2023) and generative inference in graph neural networks (Wu et al., 2024; Song et al., 2024).

The interest in analog acceleration coincides with novel proposals for "local update" algorithms, where layer activations $h$ are solutions to a minimization problem $h_\ell^* = \mathrm{argmin}_h f(h)$ (Scellier & Bengio, 2017; Stern et al., 2021; Millidge et al., 2022; Scellier et al., 2023). While costly in digital systems, stochastic analog optimizers can effectively solve $\mathrm{argmin}_h f(h)$ in minimal time and energy (Wu et al., 2024), making them suitable candidates for local-update learning implementations.

However, real-world problems are typically too large for dynamical accelerators to optimize in their entirety, requiring routines to partition and iteratively sample/optimize subspaces (Booth et al., 2017; Sharma et al., 2022; Song et al., 2024), most commonly using hybrid "large-neighborhood local search" (LNLS) frameworks (Ahuja et al., 2002; Booth et al., 2017). In hybrid LNLS, the DX is used to perform alternating sampling/minimization over within-capacity subproblems. However, hybrid

LNLS has undergone little theoretical examination. No non-asymptotic convergence bounds yet exist, limiting the appeal of hybrid LNLS compared with more well-understood digital algorithms. Moreover, the effect of algorithm hyperparameters on convergence and their interplay with device non-idealities is unclear. Models trained on one DX may require hyperparameter adjustment, if not outright device-specific retraining, prior to inference on another (He et al., 2019; Long et al., 2019). Without non-asymptotic analysis linking device variation and accelerator convergence, accelerator adaptation reduces to trial-and-error.

In this work, we provide the first explicit probabilistic convergence guarantees for hybrid LNLS algorithms in activation sampling and optimization: a crucial first step in optimizing and analyzing hybrid DX frameworks. We start by reducing hybrid LNLS to block sampling with continuous-time, Langevin diffusion-based sub-samplers, to which we can apply tools from classical sampling analysis. Two block selection rules for "block Langevin diffusion" (BLD) are examined, randomized and cyclic, using ideal (Secs. 3.2 and 3.3) and finite-variation (Sec 3.4) analog components. Under a log-Sobolev inequality (LSI), we prove that ideal accelerators converge to the target distribution exponentially fast. However, we show that finite device variation incurs a bias in $W_2$ distance, proportional to step duration and dependent on variation magnitude. We illustrate our findings with numerical experiments on a toy Gaussian sampling problem, demonstrating the effect of device variation and hyperparameter choice on convergence.

Our contributions can be summarized as follows:

1. We provide the first bounds on randomized block diffusions using explicit constants (Theorem 1), strengthening the results of Ding et al. (2020)

2. We provide completely novel bounds for cyclic block diffusions (Theorem 2) by proving a general conditional sampling lemma for Kullback-Liebler divergence (Lemma 1)

3. Using a Talagrand transportation inequality, we combine our ideal results with analysis following Raginsky et al. (2017) to provide non-asymptotic guarantees for DXs with analog non-idealities (Theorem 3), applicable to both sampling and optimization tasks.

## 2 BACKGROUND

### 2.1 RELATED WORKS

Ding & Li (2021) and Ding et al. (2021) proposed and analyzed "randomized coordinate Langevin Monte Carlo" (RCLMC) methods for sampling tasks using over and underdamped Langevin dynamics. Their methodology used Wasserstein coupling arguments akin to Dalalyan (2016), in contrast to our interpolation arguments following Vempala & Wibisono (2019). Accordingly, the authors assumed a strongly-log concave target distribution: a much stronger assumption than an LSI. Moreover, Ding et al. (2021) provided insufficient analysis for the continuous-time case, focusing primarily on the discrete RCLMC algorithm. DX algorithm analysis required continuous-time bounds with explicit constants, necessitating our contributions.

Two algorithms related to BLD garnering recent interest are "coordinate ascent variational inference" (CAVI), which performs variational inference over factorized "mean-field" distributions (Bhattacharya et al., 2023; Arnese & Lacker, 2024), and the split Gibbs sampler (SGS), which alternates sampling over problem variables with augmented priors (Vono et al., 2019; 2022). CAVI is similar to BLD, and indeed the information theoretic analysis by Lee (2022) has a similar structure to our proof of Lemma 1. SGS has been likened to ADMM in optimization Vono et al. (2022), indicating there may be an equivalence to BLD akin to classical block optimization Tibshirani (2017).

A related class of works have analyzed the accuracy of analog matrix-vector multiplication (MVM) accelerators in neural network inference (Klachko et al., 2019; Xiao et al., 2022). MVM accelerators are a restricted class of DXs minimizing $\min_{y \in \mathbb{R}^d} ||y - Wx||^2$: equivalent to performing MVM in the analog domain. Our analysis generalizes MVM analysis and is applicable in more complex analog settings such as generative sampling (Vengalam et al., 2023; Melanson et al., 2023; Wu et al., 2024).

Optimization-based convergence analyses of specific DX architectures were carried out by Erementchouk et al. (2022); Pramanik et al. (2023). Asymptotic convergence in expectation to the global minimizer was proved by Pramanik et al. (2023) in the zero-temperature limit with decreas-

ing stepsize, echoing our results in Sec. 3.4. However, neither work accounted for the effect of device variation or problem partitioning, and both focused on specific DX modalities (nonlinear electronic/optical oscillators) rather than a general model of DX behavior. Information-theoretic analysis conducted by Dambre et al. (2012); Hu et al. (2023) have bounded the asymptotic computational capabilities of DX systems, but not their probabilistic convergence.

## 2.2 DYNAMICAL ACCELERATORS

The first wave of dynamics-accelerated optimizers primarily targeted the Ising Spin Glass (ISG) Hamiltonian from statistical physics, earning the appellation "Ising Machines". The ISG Hamiltonian describes quadratic interactions between binary "spins", which can be used to solve intractable combinatorial problems (Lucas, 2014). Ising machines have been implemented using quantum spins (Ushijima-Mwesigwa et al., 2017), electronic (Wang & Roychowdhury, 2019; Albertsson & Rusu, 2023) and optical (Inagaki et al., 2016; Honjo et al., 2021) oscillator phases, resistively-coupled capacitors (Afoakwa et al., 2021), and many more besides (Mohseni et al., 2022). These initial prototypes successfully optimized binary target functions. However, recent architectures have broader applications domains: with support for non-quadratic cost functions (Sharma et al., 2023; Bashar & Shukla, 2023; Bybee et al., 2023) and continuous values (Brown et al., 2024; Wu et al., 2024; Song et al., 2024). Since these designs have moved beyond the ISG Hamiltonian, we term this broader class simply as "dynamical accelerators" (DXs).

While the physical implementation differs between DXs, several proposals can be described by a Langevin stochastic differential equation (SDE)

$$dx_t = -\nabla h(x_t, t)dt + \sqrt{2\beta(t)^{-1}}dW_t \tag{1}$$

where $x_t \triangleq x(t)$ is the system state, $dW_t$ is a Brownian noise term, $h(x, t)$ is the deterministic system potential, and $\beta(t) = 1/T(t)$ is the (potentially time dependent) inverse pseudo-temperature of the system.

$x(t)$ represents the continuous, physical degrees of freedom of the optimizer/sampler, such as capacitor voltage (Afoakwa et al., 2021) or oscillator phase (Inagaki et al., 2016; Wang & Roychowdhury, 2019). Several DX prototypes have been shown to follow forms of Equation (1), either intentionally to escape local minima (Wang & Roychowdhury, 2019; Sharma et al., 2023; Aifer et al., 2023) or unintentionally to model dynamic environment noise (Wang et al., 2013). The potential $h(x, t)$ includes the target function $f(x)$ along with optional time-dependent terms, such as a sub-harmonic injection locking potential for binarization (Wang & Roychowdhury, 2019).

DXs are also prone to static "device variation" owing to analog non-idealities. Unlike the Brownian term $dW_t$, static non-idealities are not self-averaging, and result in a biased estimate $g_\delta(x)$ of the gradient $\nabla f(x)$. In a quadratic function $f(x) = x^T W x$, for instance, the gradient estimate can be described as $g_\delta(x) = (W + W^T)x + \delta x$, where $\delta_{ij} \sim \mathcal{N}(0, \Delta^2)$ are fixed non-idealities in device components. Previous studies have examined the impact of static variation on binary optimization (Albash et al., 2019) and matrix-vector multiplication (Xiao et al., 2022), but have not extended to non-asymptotic convergence analysis for more general functions over $\mathbb{R}^d$.

## 2.3 LANGEVIN DIFFUSION

If we restrict our analysis to the time-homogeneous case where $h(x, t) = f(x)$, $\beta(t) = \beta$, the dynamics are Markovian with a constant stationary distribution

$$\pi_\beta(x) \propto e^{-\beta f(x)}.$$

The Langevin SDE

$$dx_t = -\nabla f(x_t)dt + \sqrt{2\beta^{-1}}dW_t$$

produces a continuous sample path $x(t)$ with each $x(\tau), \tau \geq 0$ acting as a random variable. The law of $x_t$, $\mu_t$ (denoted $\mu_t = \mathcal{L}(x_t)$), is described by the *Fokker-Planck* equation (FPE)

$$\partial_t \mu_t = \beta^{-1}\nabla^2 \cdot \mu_t + \nabla \cdot [\mu_t \nabla f(x_t)].$$

The Langevin SDE describes the physical evolution of $x_t$, while the FPE describes the change in the sample distribution $\mu_t$ in measure space. If $\pi_\beta$ satisfies a log-Sobolev inequality (LSI, see Sec. 3), then $\mu_t$ converges to $\pi_\beta$ exponentially fast (Theorem 1 Vempala & Wibisono, 2019).

**Theorem 0** (LD Convergence (Theorem 1 of Vempala & Wibisono (2019))). *Suppose* $\pi_\beta(x) \propto e^{-\beta f(x)}$ *satisfies an LSI with constant* $1/\gamma$. *Then the distribution* $\mu_t$ *of the Langevin diffusion at time* $t$ *satisfies*

$$\mathrm{D}_{\mathrm{KL}}(\mu_t \| \pi) \leq e^{-2\gamma\beta^{-1}t} \, \mathrm{D}_{\mathrm{KL}}(\mu_0 \| \pi) \tag{2}$$

where $\mathrm{D}_{\mathrm{KL}}(\mu_t \| \pi) \triangleq \int \mu_t(x) \log \frac{\mu_t(x)}{\pi(x)} dx \triangleq \int \mu_t \log \frac{d\mu_t}{d\pi}$ is the Kullback-Leibler (KL) divergence between two probability measures.

Recalling the Otto-Villani theorem (Theorem 1 Otto & Villani, 2000), an LSI inequality further implies a Talagrand transportation inequality

$$W_2(\mu_t, \pi) \leq \left(\frac{2}{\gamma}\right)^{1/2} \sqrt{\mathrm{D}_{\mathrm{KL}}(\mu_t \| \pi)}$$

where $W_2(\mu_t, \pi) = \inf_{\nu \in \mathcal{C}(\mu_t, \pi)} \left(\int \|x - y\|_2^2 \nu(x, y) dx dy\right)^{1/2}$ is the 2-Wasserstein distance between $\mu_t$ and $\pi$ and $\nu \in \mathcal{C}(\mu_t, \pi)$ is a *coupling* over $\mu_t$, $\pi$. Convergence in $\mathrm{D}_{\mathrm{KL}}$ under an LSI therefore implies convergence in $W_2$, allowing us to state bounds in both. Crucially for our purposes, the 2-Wasserstein distance is a metric over probability distributions, allowing use of the triangle inequality (Raginsky et al., 2017).

As $\beta \to \infty$, $\pi_\beta(x)$ concentrates around the minimizer(s) of $f$. This observation permits us to unite optimization and sampling using annealing schemes (Kirkpatrick, 1984; Chiang et al., 1987; Chak et al., 2023) which gradually increase $\beta$ to escape early local minima and (hopefully) find the global minimum, indicating a direction for future work extending our results. Previous works have also used bounds on convergence to $\pi_\beta$ at constant $\beta$ to bound optimizer hitting times Zhang et al. (2017) and expected excess risk (Raginsky et al., 2017; Xu et al., 2020; Farghly & Rebeschini, 2021; Zhang et al., 2023) in non-convex optimization.

## 3 MAIN RESULTS

### 3.1 LNLS AS BLOCK SAMPLING

DXs have a finite capacity, necessitating hybrid analog/digital algorithms to solve problems exceeding that capacity. A popular candidate for hybrid optimization/sampling is the Large-Neighborhood Local Search (LNLS) framework (Ahuja et al., 2002; Booth et al., 2017; Sharma et al., 2022; Raymond et al., 2023), where a local solver (the DX) is used to optimize/sample blocks of variables $\{B_1, B_2, ..., B_b\}$ conditioned on the rest of the problem state, illustrated in Fig. 1.

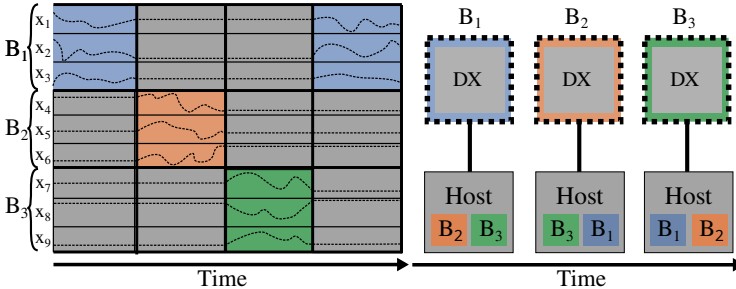

Figure 1: Illustration of the LNLS algorithm on a 3-block, 9-variable problem. **[Left]** An illustration of the variable sample paths during algorithm execution. When a block is not being actively evolved, the constituent variables remain fixed (gray). **[Right]** Logical partition of variables in an LNLS framework, where one block is being actively evolved by the DX with the others resident in digital memory. The digital host performs the control operations needed to read the block state, write back to memory, and begin the next block evolution.

We can formalize LNLS by borrowing notation from classical coordinate descent (Nesterov, 2012; Beck & Tetruashvili, 2013). We decompose the space $\mathbb{R}^d$ into disjoint blocks $\bigtimes_{i=1}^b B_i$, where

---

**Algorithm 1** Block Langevin Diffusion (BLD)

---

1: **procedure** BLD($x^0 \in \text{dom}(f)$, Decomposition $\{B_1, ..., B_b\}$, Step Size Set $\lambda \in \mathbb{R}_+^b$)
2:    **for** $k \geq 0$ **do**
3:        Choose block index $i$ (Random or Deterministic)
4:        Set $t_{k+1} = t_k + \lambda_i$
5:        Sample:
$$x(t_{k+1}) = x(t_k) - \int_{t_k}^{t_{k+1}} U_i \nabla f(x) dt + \int_{t_k}^{t_{k+1}} U_i \sqrt{2\beta^{-1}} dW_t$$
6:    **end for**
7: **end procedure**

---

each subspace $B_i$ has dimension $d_i$. LNLS frameworks perform alternating sampling from each conditional distribution $\mu_{B_i|B_1...B_b}$, where each block is chosen at random or in a deterministic order.

For simplified notation, we decompose $I^d = \sum_{i=1}^{b} U_i$ where each $U_i \in \mathbb{R}^{d \times d}$ has ones along diagonal indices corresponding to dimensions in $B_i$ and zeros elsewhere. Then $\sum_{i=1}^{b} U_i \nabla f(x) = \nabla f(x)$ and we can express the SDE for a single block $B_i$ diffusion as

$$dx = -U_i \nabla f(x) dt + U_i \sqrt{2\beta^{-1}} dW_t. \tag{3}$$

Equation (3) leaves the conditioned dimensions $\overline{B}_i \triangleq \{j \in \{1, ..., d\} : j \notin B_i\}$ invariant. Each block diffusion occurs in continuous time, but the blocks are swapped at discrete steps. Accordingly, we denote $x(t_k)$ as the iterate at time $t$ in block step $k$ and $\mu_{t_k}$ as its associated probability distribution.

When each block evolves at constant $\beta$ according to Equation (3), we can model LNLS as a block sampling algorithm, termed *Block Langevin Diffusion* (BLD), shown in Algorithm 1. BLD is a continuous-time generalization of "randomized coordinate Langevin Monte Carlo" (RCLMC) studied in Ding et al. (2021); Ding & Li (2021). By reducing LNLS to BLD, we can tractably analyze algorithm performance using well-developed tools from stochastic process analysis. As expressed, Algorithm 1 leaves open the choice of block selection. Here we consider randomized and cyclic selection rules, denoted *Randomized Block Langevin Diffusion* (RBLD) and *Cyclic Block Langevin Diffusion* (CBLD) respectively.

Throughout our analysis, we make the following assumptions on $f$.

**Assumption 1.** $f$ is continuously differentiable

**Assumption 2.** $\pi_\beta \propto \exp[-\beta f(x)]$ satisfies a log-Sobolev inequality (LSI) with $C_{\text{LSI}} = \frac{1}{\gamma}$. That is, for all distributions $\mu$ with finite second moment

$$\mathrm{D}_{\mathrm{KL}}(\mu \| \pi_\beta) \triangleq \int \mu(x) \log \frac{\mu(x)}{\pi_\beta(x)} dx \leq \frac{1}{2\gamma} \overbrace{\int \mu(x) \|\nabla \log \frac{\mu(x)}{\pi_\beta(x)}\|^2 dx}^{\mathrm{FI}(\mu \| \pi_\beta)}$$

where $\mathrm{FI}(\mu \| \pi_\beta)$ is the (relative) *Fisher information*. An LSI can hold even in non-log-concave distributions, making it a more general assumption than the strong log-concavity presumed by Ding & Li (2021). Examples include globally strongly log-concave measures with bounded regions of non-log concavity (Raginsky et al., 2017; Ma et al., 2019), high-temperature spin systems (Bauerschmidt & Bodineau, 2019) and heavy-tailed distributions which may not be *strongly* log-concave (such as measures with potentials $f$ satisfying a Kurdyka-Łojasiewicz inequality Bolte et al. (2010)).

## 3.2 RANDOMIZED BLOCK LANGEVIN DIFFUSION

In the randomized case, we select the next variable block according to a probability mass function $\phi \in \mathbb{R}^b$ with all $\phi_i > 0$. Ding et al. (2021) analyzed RCLMC using Wasserstein coupling arguments, however our analysis builds on the traditional proof of Equation (2) which relies on the *de Bruijin identity*

$$\partial_t \mathrm{D}_{\mathrm{KL}}(\mu_t \| \pi_\beta) = -\beta^{-1} \mathrm{FI}(\mu_t \| \nu)$$

which, when combined with the LSI, proves exponential convergence since $-\operatorname{FI}(\mu_t\|\nu) \leq -2\gamma \operatorname{D_{KL}}(\mu_t\|\pi_\beta)$. In the same vein, we use probabilistic arguments in Appendix C.2 to prove a de Bruijin *in*equality

$$\partial_t \operatorname{D_{KL}}(\mu_t\|\pi_\beta) \leq -\phi_{\min}\beta^{-1}\operatorname{FI}(\mu_t\|\nu)$$

where $\phi_{\min}$ is the minimum block probability in $\phi$.

By integrating and expanding the inequality, we easily obtain convergence in $\operatorname{D_{KL}}(\mu(t_k)\|\pi_\beta)$, expressed in Theorem 1. We also prove convergence for a discrete-time variant (RBLMC) in Appendix C.

**Theorem 1** (RBLD $\operatorname{D_{KL}}(\mu_{t_k}\|\pi_\beta)$ Convergence). *Let $\phi = (\phi_1, ..., \phi_b)$ be the block selection probability mass function, $\phi_i > 0$, and let $\lambda = (\lambda_1, ..., \lambda_b) \in \mathbb{R}^b$ be the sampling times for each block, $\lambda_i > 0$. For any $\pi_\beta \propto \exp[-\beta f(x)]$ satisfying Assumptions 1 and 2, and any $\beta > 0$, the sample distribution after $k$ steps of RBLD ($\mu_{t_k}$) satisfies*

$$\operatorname{D_{KL}}(\mu_{t_k}\|\pi_\beta) \leq e^{-2\gamma\beta^{-1}\phi_{\min}\lambda_{\min}k}\operatorname{D_{KL}}(\mu_0\|\pi_\beta).$$

### 3.3 CYCLIC BLOCK LANGEVIN DIFFUSION

While our randomized results tighten existing theory, real-world instances of LNLS often use cyclic orderings Sharma et al. (2022); Song et al. (2024); Wu et al. (2024), as they are more amenable to direct hardware and software optimization and are simpler to implement.

However, unlike RBLD, we cannot easily prove a "de Bruijin inequality" for CBLD. Instead, we make extensive use of the *chain lemma* for $\operatorname{D_{KL}}$

$$\operatorname{D_{KL}}(\mu\|\nu) = \mathbb{E}_{\mu_D}[\operatorname{D_{KL}}(\mu_{A|E}\|\nu_{A|E})] + \operatorname{D_{KL}}(\mu_D\|\nu_E).$$

where $A, E$ are disjoint subspaces of $\mathbb{R}^d$, $A \cup E = \mathbb{R}^d$, and $\mu, \nu$ are measures supported on $\mathbb{R}^d$ with $\mu_{A|E}$ denotes the measure over $A$ conditioned on $E = y$ for arbitrary $y$. Note that if we set $A = B_i$, $E = \overline{B}_i$, the CBLD diffusion will result in exponential contraction in $\mathbb{E}_{\mu_{\overline{B}_i}}[\operatorname{D_{KL}}(\mu_{B_i|\overline{B}_i}\|\nu_{B_i|\overline{B}_i})]$ while leaving $\operatorname{D_{KL}}(\mu_{\overline{B}_i}\|\nu_{\overline{B}_i})$ constant. CBLD then trivially results in non-increasing $\operatorname{D_{KL}}(\mu(t_k)\|\pi_\beta)$, however expressing descent across iterations is more subtle due to the sub-additivity of KL-divergence.

Taking inspiration from Beck & Tetruashvili (2013), we bound descent across $b$ steps, an entire "cycle" over the problem space, expressed in a general lemma for $\operatorname{D_{KL}}(\mu_{t_k}\|\pi_\beta)$ (proved in Appendix D).

**Lemma 1** (Cyclic $KL$ Contraction). *Let the set $C = \{C_1, ..., C_b\} \in \mathbb{R}^b_+$ satisfy $0 < C_i < 1$, and let $D_i \in \mathbb{R}$ be arbitrary constants $D_i \geq 0$ and let $\pi$ be an arbitrary distribution with finite second moment. Suppose $(\mu_0, \mu_1, ...)$ is a sequence of measures satisfying for $k \geq 1$ and $n = k \mod b$*

$$\operatorname{D_{KL}}(\mu_{t_k}\|\pi_\beta) \leq C_n \operatorname{D_{KL}}(\mu_{t_{k-1}}\|\pi) + (1 - C_n)\operatorname{D_{KL}}(\mu_{t_{k-1},\overline{B}_n}\|\pi_{\overline{B}_n}) + D_n.$$

*Then we can bound*

$$\operatorname{D_{KL}}(\mu_{t_{kb}}\|\pi) \leq C_{\max}^k \operatorname{D_{KL}}(\mu_{t_{(k-1)b}}\|\pi) + \sum_{i=1}^b D_i$$

*where $C_{\max} = \max\{C_1, ..., C_b\}$.*

Lee (2022) *lower* bounded the KL-divergence in Bayesian coordinate ascent variational inference by similarly comparing the change in $\operatorname{D_{KL}}$ across conditioned steps. However, their focus was on inference over mean-field parametric distributions rather than the broader class of LSI Gibbs measures, making Lemma 1 a stronger result.

The convergence of CBLD follows by choosing $D_i = 0$, $C_{\max} = e^{-2\gamma\beta^{-1}\lambda_{\min}}$:

**Theorem 2** (CBLD $\operatorname{D_{KL}}(\mu(t_k)\|\pi_\beta)$ Convergence). *Let $\sigma = (B_1, ..., B_b)$ be a given block permutation and let $\lambda = (\lambda_1, ..., \lambda_b) \in \mathbb{R}^d$ be the sampling times for each block, $\lambda_i > 0$. For any $\pi_\beta \propto \exp[-\beta f(x)]$ satisfying Assumptions 1 and 2, and any $\beta > 0$, the sample distribution after $kb$ steps of CBLD ($\mu_{t_{kb}}$) satisfies*

$$\operatorname{D_{KL}}(\mu_{t_{kb}}\|\pi_\beta) \leq e^{-2\gamma\beta^{-1}\lambda_{\min}k}\operatorname{D_{KL}}(\mu_0\|\pi_\beta).$$

When $D_i \neq 0$, Lemma 1 accounts for biased sampling algorithms, such as Langevin Monte Carlo (LMC). Accordingly, we combine Lemma 1 with existing LSI bounds for LMC from Vempala & Wibisono (2019); Chewi et al. (2021) to prove convergence for a discrete time "cyclic block Langevin Monte Carlo" in Appendix D.

**Observations:** For RBLD and CBLD, the convergence is limited by the shortest step duration $\lambda_{\min}$ and minimal block probability $\phi_{\min}$. For constant block sizes, the optimal choice for both CBLD and RBLD is therefore constant $\lambda_i = \lambda_j = \lambda$ and uniform $\phi_i = 1/b$. This contrasts discrete-time block optimization, where distinct step sizes/probability distributions provide advantage on ill-conditioned problems (Nesterov, 2012; Beck & Tetruashvili, 2013; Ding et al., 2021) due to the effect of varying Lipschitz constants in discretization error terms. In the case of constant $\lambda$ with uniform $\phi$, RBLD and CBLD have identical descent bounds, as we numerically demonstrate in Section 4. This considerably simplifies hyperparameter selection *for ideal devices*, reducing from $O(b)$ parameters to 1 ($\lambda$). In the following section we continue to assume a constant step duration $\lambda$ for simplicity, though future analyses may reveal more optimized step size selections for finite-variation devices.

## 3.4 FINITE VARIATION

Theorems 1 and 2 provide optimistic lower bounds for DX sampling, however a real machine will have analog errors perturbing the target function (Albash et al., 2019; Melanson et al., 2023). As a generalization of Albash et al. (2019), we model a DX with analog variation with a "perturbed" gradient oracle $g_\delta(x) : \mathbb{R}^d \to \mathbb{R}^d$, where $\delta \in \boldsymbol{D}$ denotes a fixed perturbation from arbitrary domain $\boldsymbol{D}$. Unlike stochastic optimization, which assumes that the perturbation changes with each gradient evaluation, DX perturbations are fixed for each device. To provide guarantees under device variation, we need to restrict the perturbations and functions permitted:

**Assumption 3.** For fixed $\delta \in \boldsymbol{D}$, there exist constants $M, B \geq 0$ such that for all $x \in \mathbb{R}^d$

$$\|\nabla f(x) - g_\delta(x)\|^2 \leq M^2 \|x\|^2 + B^2.$$

**Assumption 4.** $f$ is $L$-smooth and, for fixed $\delta \in \boldsymbol{D}$, $g_\delta$ is $G$-Lipschitz continuous. That is, for all $x, y \in \mathbb{R}^d$

$$\|\nabla f(x) - \nabla f(y)\| \leq L\|x - y\|,$$
$$\|g_\delta(x) - g_\delta(y)\| \leq G\|x - y\|.$$

**Assumption 5.** $f$ and $g_\delta$ are $(m, c)$-dissipative and $(\mathfrak{m}, \mathfrak{c})$-dissipative respectively, i.e., there exists positive constants $m > 0$, $c$, $\mathfrak{m}$, $\mathfrak{c} \geq 0$ such that for all $x \in \mathbb{R}^d$:

$$\langle \nabla f(x), x \rangle \geq m\|x\|^2 - c,$$
$$\langle \nabla g_\delta(x), x \rangle \geq \mathfrak{m}\|x\|^2 - \mathfrak{c}.$$

Assumption 3 limits the Euclidean distance between $\nabla f$ and $g_\delta$, with the constants $M$ and $B$ appearing in later bounds. Assumption 5 is a common assumption in analyses of stochastic gradient sampling algorithms (Raginsky et al., 2017; Li & Wang, 2022; Zhang et al., 2023). Specifically, it enables us to bound the ideal Langevin second moment $\mathbb{E}\|y(t_k)\|^2$ in the proof of Theorem 3. Assumption 4 is not directly used in our proofs, but is required for a Girsanov change of measure. Assumptions 3 and 5 both restrict the type of perturbation with Assumption 5 also limiting the magnitude. If Assumptions 4 and 5 hold for the target potential, it is reasonable to expect that they hold for the perturbed oracle as well, since DX variation typically manifests as additive or multiplicative perturbations in analog components implementing $\nabla f$ Xiao et al. (2022); Aifer et al. (2023).

Take the example of a Gaussian potential $f(x) = \frac{1}{2}x^\top \Sigma^{-1} x$ with $g_\delta(x) = \Sigma^{-1} \circ (1+\delta)x$, where $\delta \in \mathbb{R}^{d \times d}$ is a "perturbation matrix" with $\delta_{ij} \sim \mathcal{N}(0, \Delta^2)$ and $\circ$ denotes a component-wise Hadamard product. Regardless of the standard deviation $\Delta$, we satisfy Assumptions 3 and 4 with $M$ and $L$ both equal to the maximal magnitude eigenvalue of $\delta$ and $\Sigma^{-1}$ respectively with $B = 0$. However, if $\Sigma^{-1}(1+\delta)$ has negative eigenvalues there is no $\mathfrak{m} > 0$ satisfying Assumption 5, placing an upper limit on the perturbation strength.

**Assumption 6.** The density of the initial law $\mu_0$ satisfies

$$\kappa_0 \triangleq \log \int_{\mathbb{R}^d} e^{\|w\|^2} d\mu_0 < \infty.$$

In practice, dynamical accelerators typically operate over bounded domains, such as the unit hypercube (Afoakwa et al., 2021) or unit circle (Wang & Roychowdhury, 2019; Inagaki et al., 2016), hence the iterate magnitude is bounded in any case. However, bounding over the entire space would provide insufficiently tight upper bounds and our methodology assumes that the measures are supported on $\mathbb{R}^d$. We leave consideration of domains with bounded support to future work, potentially applying methods from reflected Langevin diffusion theory (Bubeck et al., 2018) or projected differential analysis (Cherukuri et al., 2016).

We begin by stating the following bound on the distance between the measures of ideal and perturbed BLD, proved in Appendix E:

**Lemma 2** (Finite Variation Block Langevin $W_2$ Distance). *Let $x(t)$, $y(t)$ be non-ideal and ideal block Langevin processes respectively with associated distributions $\mu_t, \nu_t$. For any $\pi_\beta \propto \exp[-\beta f(x)]$ satisfying 1, 2, and 5 with $\beta > \frac{2}{m}$, $g_\delta(x)$ satisfying Assumption 3, $\mu_0$ satisfying Assumption 6, and $k\lambda > 1$ after $kb$ iterations of BLD*

$$W_2(\mu_{t_{kb}}, \nu_{t_{kb}}) \leq \sqrt{C_0 \left[ (C_1 + \sqrt{C_1}) + (C_2 + \sqrt{C_2})\sqrt{\lambda} \right] k\lambda}$$

*where $C_0$, $C_1$, and $C_2$ are given in Appendix E.*

From previous discussions, setting $\phi_i = 1/b$, $\lambda_i = \lambda$ unifies the bounds for RBLD and CBLD. In this regime, we can prove the following statement as a simple consequence of the triangle inequality $W_2(\mu, \nu) \leq W_2(\mu, \eta) + W_2(\eta, \nu)$ and the Otto-Villani theorem

**Theorem 3** (Finite-Variation BLD $W_2$ Convergence).

$$W_2(\mu_{t_{bk}}, \pi_\beta) \leq \left( \frac{2}{\gamma} \right)^{1/2} e^{-\gamma\beta^{-1}\lambda k} \sqrt{D_{KL}(\mu_0 \| \pi_\beta)} + \sqrt{C_0 \left[ (C_1 + \sqrt{C_1}) + (C_2 + \sqrt{C_2})\sqrt{\lambda} \right] k\lambda}.$$

Following Raginsky et al. (2017), if we choose $k\lambda = \frac{\beta}{\gamma} \log \frac{2\sqrt{2 D_{KL}(\mu_0 \| \pi_\beta)}}{\varepsilon\sqrt{\gamma}}$ and set $\lambda \leq (\varepsilon\gamma)^4 \left( \beta \log \left[ 2\sqrt{2 D_{KL}(\mu_0 \| \pi_\beta)}/(\sqrt{\gamma}\varepsilon) \right] \right)^{-4}$, we have

$$W_2(\mu_{t_{bk}}, \pi_\beta) \leq \frac{\varepsilon}{2} + \sqrt{C_0} \left[ \sqrt{C_1 + \sqrt{C_1}} \frac{\beta}{\gamma} \log \frac{2\sqrt{2 D_{KL}(\mu_0 \| \pi_\beta)}}{\varepsilon\sqrt{\gamma}} + \varepsilon\sqrt{C_2 + \sqrt{C_2}} \right]. \quad (4)$$

We thereby obtain a total bound on the Wasserstein error $\mathcal{O}(\log \frac{1}{\varepsilon} + \varepsilon)$ for arbitrary $\varepsilon > 0$.

**Observations:** Similar to discrete-time LMC, our Wasserstein bound has a non-zero lower bound with respect to $\varepsilon$: non-ideal devices introduce *bias*. Unlike discrete LMC, the bias in Equation (4) does not result from a forward-flow discretization (Wibisono, 2018; Chewi et al., 2021). Instead, the constants $C_0, C_1, C_2$ are solely due to finite analog variation. For $M = 0$, $B = 0$, we recover exponential, unbiased convergence in $W_2$. However, akin to LMC, practitioners can select the step size $\lambda$ and the injected noise $\beta$ to control the bias. Higher temperatures (lower $\beta$) result in a lower bias, as expected from the application of a noisy channel in measure space. Moreover, DX users/designers typically characterize $M, B$ during device calibration: simultaneously lowering the impact of analog non-ideality and allowing for a rough bound on the distribution bias (See Section C.2.a of Melanson et al., 2023).

A Wasserstein bound suffices as a performance guarantee in sampling tasks such as Boltzmann machine inference (Hinton et al., 2006) or statistical physics simulation (Hamerly et al., 2019; Ng et al., 2022; Inaba et al., 2023). For optimization, assuming quadratic function growth with $\beta \geq \frac{2}{m}$ and a dissapative gradient oracle (Assumption 5) allows the use of a continuity inequality (Lemma 6 of Raginsky et al., 2017) and second moment bound (Proposition 11 of Raginsky et al., 2017) to bound $\mathbb{E}_{\mu_{t_k}}[f(x)] - \mathbb{E}_{\pi_\beta}[f(x)]$ and $\mathbb{E}_{\pi_\beta}[f(x)] - \min_{x \in \mathbb{R}^d} f(x)$ respectively

$$\mathbb{E}_{\mu_{t_{bk}}}[f(x)] - \mathbb{E}_{\pi_\beta}[f(x)] \leq (M\sigma + B)W_2(\mu_{t_{bk}}, \pi_\beta), \quad (5)$$

$$\mathbb{E}_{\pi_\beta}[f(x)] - \min_{x \in \mathbb{R}^d} f(x) \leq \frac{d}{2\beta} \log \left( \frac{eL}{m} \left( \frac{c\beta}{d} + 1 \right) \right) \quad (6)$$

where $\sigma^2 = \max\{\mathbb{E}_{\mu_{t_{bk}}}[x^2], \mathbb{E}_{\pi_\beta}[x^2]\}$ (given in Appendix E). Combining Equations (5) and (6), we obtain

$$\mathbb{E}_{\mu_{t_{bk}}}[f(x)] - \min_{x \in \mathbb{R}^d} f(x) \leq \frac{d}{2\beta} \log\left(\frac{eL}{m}\left(\frac{c\beta}{d}+1\right)\right) + (M\sigma + B)W_2(\mu_{t_{bk}}, \pi_\beta)$$
$$= \mathcal{O}(\frac{d}{\beta}\log\beta d + (M+B)(\varepsilon + \log\varepsilon^{-1})).$$

Controlling the first term requires increasing $\beta$ (lower injected noise) in tandem with problem dimension. Conversely, controlling the second term requires $\lambda kb \propto \beta$, $\lambda \propto \frac{1}{\beta^4}$, i.e., more iterations with lower step duration with increasing $\beta$. In digital algorithms, we are free to choose $\lambda$ arbitrarily small to meet given precision requirements (though the program convergence may be impracticably slow). Dynamical accelerators typically have a lower bound on $\lambda$ (e.g. a digital clock), translating into an effective upper bound on $\beta$.

## 4 NUMERICAL EXPERIMENTS

As an illustrative example, we simulated CBLD and RBLD behavior for $d = 50$ Gaussian sampling with $\beta = 1$ and uniform $\lambda_i = \lambda$[1]. Gaussian distributions permit closed-form solutions for $D_{KL}(\mathcal{N}(u_1, \Sigma_1) \| \mathcal{N}(u_2, \Sigma_2))$, allowing for a quantitative estimate of convergence. Moreover, several proposed use cases for DXs rely on Gaussian sampling, including matrix inversion (Aifer et al., 2023) and uncertainty quantification (Melanson et al., 2023). Other works have also proposed using DXs to optimize strongly-convex functions of the form $f(x) = (x-u)^\top W(x-u)$ (Wu et al., 2024; Song et al., 2024). As discussed in the preceding section, our bounds provide expected function gap guarantees from sampling $\pi = \mathcal{N}(u, 2\beta^{-1}W^{-1})$, where optimization would occur in the $\beta \to \infty$ limit. Appendix A gives more experimental details.

Fig. 2a shows the convergence in $D_{KL}$ for the estimated distribution $\mu_t$ with block counts $b \in \{1, 2, 5, 10\}$ versus simulated time. Note that we are only comparing the convergence *relative to ideal LD* ($b = 1$) rather than the exact rate or $D_{KL}$ value (see Appendix A). The block methods converge slower than the full-gradient process, with the rate of convergence decreasing with more blocks. By comparing the exponents in Theorems 0 ($-2\gamma\beta^{-1}t$) and 1 ($-2\gamma\beta^{-1}\lambda\phi_{\min}k$), we note that choosing $k = \frac{t}{\phi_{\min}\lambda}$ makes the two contractions equal. Similarly, the exponent in Theorem 2 for a contraction in $b$ itera-

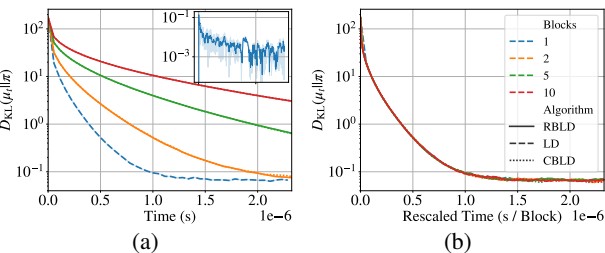

(a)                 (b)

Figure 2: Convergence in $D_{KL}$ for varying block counts $b$ with for $b-$BCLD and $b-$RCLD **(a)** versus simulated time and **(b)** versus cycles $kb$. The inset plot in **(a)** shows the absolute difference between the RBLD and CBLD $D_{KL}$ values averaged over each block count.

tions ($-2\gamma\beta^{-1}\lambda k$) suggests the choice $kb = \frac{t}{\lambda}$ iterations. Since $\lambda$ is the amount of time, spent per block, this suggests that equating the total time per block should result in roughly equivalent contractions. Fig. 2b confirms this prediction by showing the same data after rescaling the x-axis by $b$ for each method, with the curves converging within sampling error. Block sampling therefore incurs an $O(b)$ slowdown in real time, similar to coordinate methods in optimization (without accounting for smoothness (Wright, 2015)).

---

[1]Code and data can be found at `https://github.com/ur-acal/BlockLangevin`.

Our testing showed that all step sizes lead to the same convergence rate with respect to time (not shown). However, larger step sizes lead to larger decay w.r.t. whole problem *cycles* ($b$ iterations), as shown in Fig. 3a. This reinforces the importance of *per block* sampling time for continuous coordinate methods. Finally, we perturb the similarity matrix $\Sigma^{-1}$ with component-wise variation $\tilde{\Sigma}_{ij} = \Sigma_{ij}(1 + \delta_{ij})$, $\delta_{ij} \sim \mathcal{N}(0, \Delta)$. Fig. 3b shows the impact on $D_{\mathrm{KL}}$ convergence with increasing perturbation strength. At $\Delta = 0.6$, $\Sigma$ is no longer positive-definite, causing the iterate to diverge (not shown on plot). $\Delta \in \{0.1, 0.2, 0.3, 0.4, 0.5\}$ satisfied Assumption 5 and were therefore stable (though biased), but $\Delta = 0.6$ did not, in line with discussion in the preceding section.

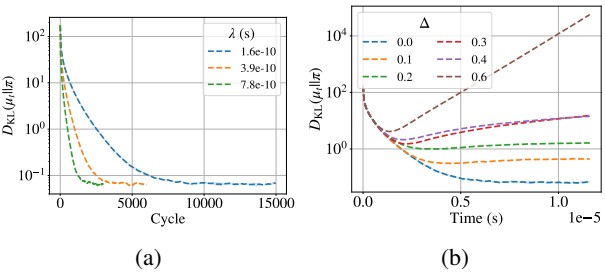

(a)           (b)

Figure 3: BCLD $D_{\mathrm{KL}}$ convergence **(a)** versus whole-problem cycles with varying block duration $\lambda$ and **(b)** versus simulated time with varying multiplicative Gaussian perturbations.

## 5   CONCLUSION

**Findings:** In this work, we prove novel bounds for Large Neighborhood Local Search (LNLS) sampling algorithms leveraging continuous Langevin diffusion, providing valuable information to device designers, potential DX adopters, and future analyses. Specifically, we

1. prove novel non-asymptotic convergence bounds for randomized and cyclic block selection strategies, finding that both methods produce identical convergence rates in KL-divergence.

2. provide probabilistic convergence bounds for LNLS sampling using non-ideal analog devices, with biases expressed using practically measurable/estimable constants

3. validate our theoretical results by numerical simulation, demonstrating ① the expected equivalence of cyclic and randomized strategies and ② the expected dependence of convergence rates on algorithm/problem parameters.

**Limitations:** Assumptions 2 and 5 provide useful bounds for many ML and optimization problems over continuous domains. However, DX applications include discrete choice problems and/or significantly non-convex potentials, such as mixed integer programming. Future bounds necessarily involve more general assumptions than the $\gamma$-LSI class considered here. Analog accelerators also typically use low-precision ($< 8b$) DACs and ADCs for input/output (Xiao et al., 2022), making studies of quantizated convergence/expected function gap critical for real-world applications.

A technical limitation is the assumption of support over $\mathbb{R}^d$. DXs operate in bounded subspaces such as the unit circle (Inagaki et al., 2016) or hypercube (Afoakwa et al., 2021). Future work applying projected differential analysis (Cherukuri et al., 2016; Bubeck et al., 2018) is therefore needed.

**Directions for Future Work:** In this work, we focused on the "inference" stage where the coupling parameters are fixed. For training, current usages of DX systems update model weights in discrete time, either by the digital controller (Song et al., 2024) or by a discrete step in the analog domain (Vengalam et al., 2023). It will be a boon to DX research to further develop theories for the training regime where model weights become the end goal for dynamics.

Finally, this work focuses on a simplified LNLS framework, relevant to the dynamics-accelerated LNLS popular in literature (Sharma et al., 2022; Raymond et al., 2023; Wu et al., 2024). It leaves open the question whether additional digital steps, such as a Metropolis-Hastings filter or replica exchange, could improve the non-asymptotic accuracy or convergence rate.

## ACKNOWLEDGMENTS

This work was supported in part by NSF under Awards No. 2231036 and No. 2233378; and by DARPA under contract No. FA8650-23-C-7312. Our thanks go to Prof. Jiaming Liang for several fruitful discussions and helpful feedback on an early draft of this work.

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

SUPPLEMENTARY MATERIALS

Here we provide proofs and explanations of experimental methods. Additionally, we apply our analysis to bound $D_{KL}$ for discrete-time variants of RBLD and CBLD (RBLMC and CBLMC). For clarity of notation, we omit the $\beta$ subscript when referring to the target distribution $\pi \triangleq \pi_\beta$.

## A  EXPERIMENTAL METHODS

### A.1  DIFFUSION SIMULATION

The purpose of our experiments is a "proof-of-concept", hence we focus on demonstrating ① the expected performance equivalence between RBLD and CBLD, ② the significance of stepsize choice, and ③ the impact of stochastic perturbations.

We simulate Langevin SDEs using an Euler-Maruyama integration scheme with a time step size of $1.6 \times 10^{-11}$ seconds. Block diffusions are simulated for a fixed number of steps, then the block is switched either cyclically or randomly, depending on the algorithm. All code is implemented in PyTorch and was run on a desktop system using an i9-13900k with 64 GB of RAM and an RTX 4090 GPU. Our simulator, plotting code, and data is publicly available at `https://github.com/ur-acal/BlockLangevin`.

We take the resistively coupled BRIM architecture from Afoakwa et al. (2021) with the Langevin perturbations proposed by Sharma et al. (2023) as our baseline DX. The BRIM architecture is more easily extensible to general classes of real-valued functions (Sharma et al., 2023; Song et al., 2024; Wu et al., 2024) than oscillator-based DXs (Wang & Roychowdhury, 2019; Inagaki et al., 2016), motivating the selection.

We model the device using $310\,\mathrm{kOhm}$ resistors and $50\,\mathrm{fF}$ capacitors, leading to an RC time constant of $1.55 \times 10^{-8}\,\mathrm{s}$ and an effective step size of $1.55 \times 10^{-11}\,\mathrm{s}$, which we use to plot total estimated DX time. These circuit parameters are comparable to those proposed in literature (Afoakwa et al., 2021; Zhang et al., 2022), however different device parameters will simply rescale the x-axis.

### A.2  TARGET POTENTIAL

As stated in the main text, we choose a Gaussian target measure to obtain a direct estimate of convergence rather than using proxy statistical observables, as done in Ding & Li (2021). The $d = 50$ Gaussian used to produce Fig. 3 was generated using the following procedure:

1. Generate a $50 \times 50$ matrix $\Sigma_\pi^{-1}$ with elements $\sim \mathrm{Unif}[-5, 5]$
2. Make the matrix symmetric by setting $\Sigma_\pi^{-1} = \frac{1}{2}(\Sigma_\pi^{-1} + (\Sigma_\pi^{-1})^\top)$
3. To make $\Sigma_\pi$ positive definite, set $\Sigma_\pi^{-1} = \Sigma^{-1} + 1.2\lambda_{\min}I_{50}$

The resulting matrix is symmetric and positive-definite, making it a valid similarity matrix. We then invert $\Sigma_\pi^{-1}$ to obtain the target covariance matrix $\Sigma_\pi$. We choose $[-5, 5]$ as the distribution to test a larger range of perturbation strengths $\Delta \in [0.1, 0.4]$, as the $W_2$ diverged much earlier ($\Delta < 0.2$) with a uniform $[-1, 1]$ distribution.

As our focus is sampling rather than optimization, we set $\beta = 1$ for simplicity. We also assume the Gaussian mean is zero, making the target distribution

$$\pi(x) \propto e^{-\frac{1}{2}x^\top \Sigma_\pi^{-1} x}.$$

### A.3  SAMPLING PROCEDURE

We randomly initialize $N = 10^4$ states and evolve them in parallel with equivalent block selections. Every 30 iterations (time $t$) we compute the empirical covariance matrix for time $\Sigma_t$ and the empirical mean $u_t$ to obtain the estimated $\mu_{t,Est} = \mathcal{N}(u_t, \Sigma_t)$. We then compute the similarity to the target Gaussian. For completeness we compute both $W_2$ and $D_{KL}$ using

$$W_2^2(\mu_{t,Est}, \mathcal{N}(0, \Sigma_\pi)) = ||\overline{x}_t||^2 + \mathrm{Tr}\left[\Sigma_t + \Sigma_\pi - 2\sqrt{\sqrt{\Sigma_\pi}\Sigma_t\sqrt{\Sigma_\pi}}\right]$$

for $W_2$ and a PyTorch library function for $D_{KL}$, which computes

$$D_{KL}(\mu_{t,Est}\|\mathcal{N}(0,\Sigma_\pi)) = \frac{1}{2}\left[\log\frac{\det\Sigma_\pi}{\det\Sigma_t} - d + \text{Tr}[\Sigma_\pi^{-1}\Sigma_t] + u_t^T\Sigma_\pi^{-1}u_t\right].$$

## B    REVIEW OF LANGEVIN DYNAMICS

In this section we review the traditional proof of Equation (2) as presented by Vempala & Wibisono (2019); Chewi (2024). Proofs in succeeding sections follow similar processes, making a brief review useful for establishing context.

Recall that the LD SDE is given by

$$dx = -\nabla f(x)dt + \sqrt{2\beta^{-1}}dW_t. \tag{7}$$

The Fokker-Planck equation (FPE) for Equation (7) is given by (Vempala & Wibisono, 2019; Jordan et al., 1998)

$$\frac{\partial\mu_t}{\partial t} = \beta^{-1}\nabla^2\cdot\mu_t + \nabla\cdot[\mu_t\nabla f(x)]$$

where $\mu_t$ is the law of $x(t)$. For convenience, we will abbreviate $\frac{\partial\mu_t}{\partial t}$ as $\partial_t\mu_t$.

Note that the right hand side of the FPE can be equivalently expressed as

$$\begin{aligned}
&\beta^{-1}\nabla^2\cdot\mu_t + \nabla\cdot[\mu_t\nabla f(x)]\\
&= \beta^{-1}\nabla\cdot[\nabla\mu_t + \beta\mu_t\nabla f(x)]\\
&= \beta^{-1}\nabla\cdot[\mu_t\nabla\ln\mu_t - \mu_t\nabla\ln\pi_\beta]\\
&= \beta^{-1}\nabla\cdot[\mu_t\nabla\ln\frac{\mu_t}{\pi_\beta}].
\end{aligned} \tag{8}$$

The time derivative of the KL-divergence is given by

$$\partial_t D_{KL}(\mu_t\|\pi_\beta) = \partial_t\int\mu_t(x)\ln\frac{\mu_t(x)}{\pi_\beta(x)}dx$$

$$= \int\left[[\partial_t\mu_t(x)]\ln\frac{\mu_t(x)}{\pi_\beta(x)} + \mu_t(x)\partial_t\ln\frac{\mu_t(x)}{\pi_\beta(x)}\right]dx$$

The second term is equal to zero, since

$$\int\mu_t(x)\partial_t\ln\frac{\mu_t(x)}{\pi_\beta(x)}dx = \int\partial_t\mu_t(x)dx = \partial_t\int\mu_t(x)dx = 0.$$

Then, applying Equation (8) and integrating by parts

$$\partial_t D_{KL}(\mu_t\|\pi_\beta) = \int[\partial_t\mu_t(x)]\ln\frac{\mu_t(x)}{\pi_\beta(x)}dx$$

$$= \int\beta^{-1}\nabla\cdot[\mu_t(x)\nabla\ln\frac{\mu_t(x)}{\pi_\beta(x)}]\ln\frac{\mu_t(x)}{\pi_\beta(x)}dx$$

$$= -\int\beta^{-1}\mu_t\left\langle\nabla\ln\frac{\mu_t(x)}{\pi_\beta(x)},\nabla\ln\frac{\mu_t(x)}{\pi_\beta(x)}dx\right\rangle$$

$$= -\int\beta^{-1}\mu_t\|\nabla\ln\frac{\mu_t(x)}{\pi_\beta(x)}\|^2dx = -\beta^{-1}FI(\mu_t\|\pi_\beta)$$

where $FI(\mu_t\|\pi_\beta)$ is the relative Fisher information of $\mu_t$ relative to $\pi_\beta$.

If $\pi_\beta(x)\propto e^{-\beta f(x)}$ satisfies a log-Sobolev inequality then

$$D_{KL}(\mu_t\|\pi_\beta)\leq\frac{1}{2\gamma}FI(\mu_t\|\pi_\beta).$$

Combining the LSI with Equation B, we obtain

$$\partial_t D_{KL}(\mu_t\|\pi_\beta) = -\beta^{-1}FI(\mu_t\|\pi_\beta)\leq -2\gamma\beta^{-1}D_{KL}(\mu_t\|\pi_\beta)$$

which implies exponential convergence of $D_{KL}(\mu_t\|\pi_\beta)$

$$D_{KL}(\mu_t\|\pi_\beta) = e^{-2\gamma\beta^{-1}t}D_{KL}(\mu_0\|\pi_\beta).$$

## C  RANDOMIZED BLOCK LANGEVIN DIFFUSION (RBLD)

In this section we provide proofs relating to Randomized Block Langevin Diffusion (RBLD, the focus of the main text) and a time-discretized version, Randomized Block Langevin Monte Carlo (RBLMC). RBLMC was previously introduced in Ding et al. (2021) as a coordinate-wise scheme, however we examine block partitions. Moreover, our results using $\gamma$-LSI target measures are more general than the strongly log-concave convergence results given in that work. For simplicity, we use the shorthand $x_t \triangleq x(t)$ throughout.

Algorithm 2 gives the structure of RBLD/RBLMC sampling, where $\phi = \{\phi_1, ..., \phi_b\}$ is a discrete probability mass function over coordinate block indices.

### C.1  CONTINUOUS TIME ITERATION

---

**Algorithm 2** Randomized Block Langevin Dynamics (RBLD)

---

1: **procedure** RBLD($x_0 \in \text{dom}(f)$, Block Distribution $\phi$ over $\{B_1, ..., B_b\}$, Step Size Set $\lambda \in \mathbb{R}_+^b$)
2:     **for** $k \geq 0$ **do**
3:         Choose $i \sim \phi$ and set $t_{k+1} = t_k + \lambda_i$
4:         Sample:

$$x_{t_{k+1}} = x_{t_k} - \int_{t_k}^{t_{k+1}} U_i \nabla f(x) dt + \int_{t_k}^{t_{k+1}} U_i \sqrt{2\beta^{-1}} dW_t$$

5:     **end for**
6: **end procedure**

---

We first consider the case when each diffusion occurs in continuous time. For a single iteration, we can formulate the evolution of the system by the following Itô SDE:

$$dx = -U_k \nabla \left( f(x) dt + \sqrt{2\beta^{-1}} dW_t \right)$$

To prove continuous-time descent in $KL$-divergence, we combine standard Langevin gradient flow arguments with methodology inspired by Ref. Vempala & Wibisono (2019) when considering expectation terms.

### C.2  FOKKER-PLANCK EQUATION

Let $\mu_t$ be the law of $x_t$, and let $\mu_{t|0}$ be the measure jointly conditioned ① on the state at time 0 and ② the choice of block $B_k$. Within a single step, $\mu_{t|0}$ will obey the Fokker-Planck continuity equation

$$\partial_t \mu_{t|0} = \text{Tr}[U_k \beta^{-1} \nabla^2 \mu_{t|0}] + \nabla \cdot (\mu_{t|0} U_k \nabla f(x_t)).$$

If we were tracking the diffusion over a single block, we would take expectation over the starting state $x_0$ while conditioning on the block index. However, as discussed in the main text, we take a "meta-Eulerian" perspective. Instead of tracking one block diffusion, our approach finds the average behavior of an ensemble of diffusion processes, each independently sampling their blocks according to $\phi$. We therefore take expectation over both $x_0$ and $B_k$ to derive the change in the "ensemble" measure $\mu_t$.

Therefore we have

$$\partial_t \mu_t = Tr[\beta^{-1} U_\phi \nabla^2 \mu_t] + \mathbb{E}[\nabla \cdot (\mu_{t|0} U_k \nabla f(x))].$$

Where we have defined $U_\phi \triangleq (\phi_1 U_1, ..., \phi_b U_b) \in \mathbb{R}^{d \times d}$.

Let $\nu$ be the joint law of $(x_0, B_k)$. Note that

$$
\begin{aligned}
\mu_t(x_t|x_0, B_k)\nu(x_0, B_k) &= \mu_t(x_t)\nu(x_0, B_k|x_t) \\
&= \mu_t(x_t)\nu(x_0|x_t, B_k)\nu(B_k|x) \\
&= \mu_t(x_t)\nu(x_0|x_t, B_k)\nu(B_k) \\
&= \mu_t(x_t)\nu(x_0|x_t, B_k)\phi_k.
\end{aligned}
$$

Then we can express the second term as

$$
\begin{aligned}
\mathbb{E}[\nabla \cdot (\mu_{t|0}U_k\nabla f(x_t))] &= \nabla \cdot \left(\sum_{i=1}^{b}\int \mu_t(x_t|x_0, i)U_k\nabla f(x_t)\nu(x_0, i)dx_0\right) \\
&= \nabla \cdot \left(\sum_{i=1}^{b}\phi_i\int \mu_t(x_t)U_i\nabla f(x_t)\nu(x_0|x_t, i)dx_0\right) \\
&= \nabla \cdot (\mu_t(x_t)U_\phi\nabla f(x_t))
\end{aligned}
$$

since

$$
\sum_{i=1}^{b}\phi_i U_i\nabla f(x_t) = U_\phi\nabla f(x_t).
$$

Therefore, the FPE of the "meta-Eulerian" RBLD process is

$$
\partial_t\mu_t = \mathrm{Tr}[\beta^{-1}U_\phi\nabla^2\mu_t] + \nabla \cdot (\mu_t U_\phi\nabla f(x_t)).
$$

Note that the we can use the identity $\nabla f(x) = \beta^{-1}\nabla\log\pi$ to re-express the FPE as

$$
\partial_t\mu_t = \nabla \cdot \left(\beta^{-1}\mu_t U_\phi\nabla\log\frac{\mu_t}{\pi}\right). \tag{9}
$$

### C.3 $KL$-DIVERGENCE CONTRACTION

**Lemma 3.**
$$
\mathrm{D}_{\mathrm{KL}}(\mu_t\|\pi) \le \mathrm{D}_{\mathrm{KL}}(\mu_0\|\pi)e^{-2\beta^{-1}\gamma\lambda_{\min}\phi_{\min}}.
$$

*Proof.* The proof follows conventional analyses of Langevin diffusion processes, e.g., see Vempala & Wibisono (2019); Chewi (2024); Chewi et al. (2021). However, we complete the proof anew for completeness, as well as to show the differences with baseline LD.

With the time evolution of the measure, we can now express the time evolution of the KL-divergence

$$
\begin{aligned}
\partial_t \mathrm{D}_{\mathrm{KL}}(\mu_t\|\pi) &= \partial_t\int \mu_t(x)\log\frac{\mu_t(x)}{\pi(x)}dx \\
&= \int \partial_t[\mu_t(x)\log\frac{\mu_t(x)}{\pi(x)}]dx \\
&= -\int \partial_t\mu_t(x)\log\frac{\mu_t(x)}{\pi(x)}dx + \overbrace{\int \partial_t\mu_t(x)dx}^{=0}
\end{aligned}
$$

where the second term is equal to zero since

$$
\int \partial_t\mu_t(x)dx = \partial_t\int \mu_t(x)dx = \partial_t[1] = 0.
$$

Using Eqn. (9), we then have

$$
\partial_t \mathrm{D}_{\mathrm{KL}}(\mu_t\|\pi) = \int\left\{\nabla \cdot \left(\beta^{-1}\mu_t U_\phi\nabla\log\frac{\mu_t}{\pi}\right)\right\}\log\frac{\mu_t(x)}{\pi(x)}dx.
$$

Through integration by parts, we obtain

$$
\begin{aligned}
\partial_t \mathrm{D}_{\mathrm{KL}}(\mu_t\|\pi) &= -\beta^{-1}\int\left\langle U_\phi\mu_t\nabla\log\frac{\mu_t}{\pi}, \nabla\log\frac{\mu_t(x)}{\pi(x)}\right\rangle dx \\
&= -\beta^{-1}\mathbb{E}_{\mu_t}\left[\left\langle U_\phi\nabla\log\frac{\mu_t}{\pi}, \nabla\log\frac{\mu_t(x)}{\pi(x)}\right\rangle\right].
\end{aligned}
$$

$U_\phi$ is positive-definite with minimum eigenvalue $\phi_{\min}$, therefore

$$\partial_t \mathrm{D_{KL}}(\mu_t\|\pi) = -\beta^{-1}\mathbb{E}_{\mu_t}\left[\left\langle U_\phi \nabla \log \frac{\mu_t}{\pi}, \nabla \log \frac{\mu_t(x)}{\pi(x)}\right\rangle\right] \le -\beta^{-1}\phi_{\min}\mathbb{E}_{\mu_t}\left[\left\|\nabla \log \frac{\mu_t}{\pi}\right\|^2\right]$$

$$= -\beta^{-1}\phi_{\min}FI(\mu_t\|\pi) \le -2\beta^{-1}\gamma\phi_{\min}\mathrm{D_{KL}}(\mu_t\|\pi)$$

where the last inequality utilizes the $\gamma$-LSI. Here we highlight a principle difference between LD and RBLD analysis. In LD, we have the "de Brujin *identity*"

$$\partial_t \mathrm{D_{KL}}(\mu_t\|\pi) = -2\beta^{-1}\gamma FI(\mu_t\|\pi).$$

However, for RBLD we have a "de Brujin *in*equality"

$$\partial_t \mathrm{D_{KL}}(\mu_t\|\pi) \le -\beta^{-1}\gamma\phi_{\min}FI(\mu_t\|\pi).$$

We now integrate up to $\lambda_k$. Since this step size depends on the choice of $k$, we take expectation of $\mathrm{D_{KL}}(\mu_{\lambda_i}\|\pi)$ where $t_k = \sum_{i=1}^k \lambda_k$

$$\mathbb{E}[\mathrm{D_{KL}}(\mu_k\|\pi)] \le \mathbb{E}[e^{-2\gamma\beta^{-1}\phi_{\min}\lambda_i}]\,\mathrm{D_{KL}}(\mu_{k-1}\|\pi)$$

or deterministically

$$\mathrm{D_{KL}}(\mu_k\|\pi) \le e^{-2\gamma\beta^{-1}\phi_{\min}\lambda_{\min}}\,\mathrm{D_{KL}}(\mu_{k-1}\|\pi).$$

Expanding the inequality $k$ times yields the result. $\qquad\square$

### C.4 RCLMC: EULER-MARUYAMA DISCRETIZATION

We now extend our analysis to discrete-time Randomized Block Langevin Monte Carlo (RBLMC), shown in Algorithm. 3. While the continuous-time diffusion can be implemented on dynamical

---

**Algorithm 3** Randomized Block Langevin Monte Carlo (RBLMC)

---

1: **procedure** RBLMC($x_0 \in \mathrm{dom}(f)$, Block Distribution $\phi$ over $\{B_1, ..., B_b\}$, Step Size Set $\lambda \in \mathbb{R}_+^b$)
2:     **for** $k \ge 0$ **do**
3:         Choose $i \sim \phi$, sample $\xi^k \sim \mathcal{N}(0, I^d)$
4:         Set:

$$x^{k+1} = x^k - \lambda_i U_i \nabla f(x^k) + U_i \sqrt{2\lambda_i}\xi^k$$

5:     **end for**
6: **end procedure**

---

hardware, digital applications require an error bound in the discrete-setting. The following derivation closely follows the methods of Vempala & Wibisono (2019) by modeling the divergence of the discrete scheme from a continuous-time interpolation. To simplify constant terms, we take $\beta = 1$ for this section.

We now consider the SDE

$$dx = U_k[-\nabla f(x_0)dt + \sqrt{2}dW_t]$$

where $x_0$ is the initial state. The SDE has the solution

$$x_t = x_0 + U_k[-\nabla f(x_0)t + \sqrt{2t}\xi_t]$$

for $t \in [0, \lambda_n]$ and $\xi_t \sim \mathcal{N}(0, I^{d_i})$. Conditioned on the initial state $x_0$ and the choice of $i$, we have the FPE

$$\partial_t \mu_{t|k,x_0} = \nabla^2 \cdot \mu_{t|k,x_0} + \nabla \cdot \mu_{t|k,x_0} U_i \nabla f(x_0).$$

Taking expectation over both sides (as previously) yields

$$\partial_t \mu = \mathrm{Tr}[U_\phi \nabla^2 \mu_{t|k,x_0}] + \nabla \cdot \mathbb{E}[\mu_{t|k,x_0} U_i \nabla f(x_0)].$$

Again noting that the choice of block and the initial state $x_0$ are independent, we can express the expectation as

$$\mathbb{E}[\mu_{t|k,x_0} U_k \nabla f(x_0)] = \sum_{i=1}^{b} \phi_i \int \mu(x_t|i, x_0)\nu(x_0) U_i \nabla f(x_0) dx_0.$$

Note that while $x_0$ and $\phi_i$ are independent random variables, they are not independent when conditioned on $x_t$. We then have

$$\begin{aligned}
\phi_i \mu(x_t|i, x_0)\nu(x_0) &= \mu(x_t|i, x_0)\nu(x_0, i) \\
&= \mu(x_t)\nu(x_0, i|x_t) \\
&= \mu(x_t)\nu(x_0|x_t, i)\phi_{i|x_t} \\
&= \mu(x_t)\nu(x_0|x_t, i)\phi_i.
\end{aligned}$$

Then

$$\begin{aligned}
\sum_{i=1}^{b} \phi_i \int \mu(x_t|i, x_0)\nu(x_0) U_i \nabla f(x_0) dx_0 &= \sum_{i=1}^{b} \phi_i \int \mu(x_t)\nu(x_0|x_t, i) U_i \nabla f(x_0) dx_0 \\
&= \mu(x_t) \int \nu(x_0|x_t) U_\phi \nabla f(x_0) dx_0 \\
&= \mu_t U_\phi \mathbb{E}[\nabla f(x_0)].
\end{aligned}$$

We then have the following FPE

$$\begin{aligned}
\partial_t \mu_t &= \text{Tr}[U_\phi \nabla^2 \mu_t] + \nabla \cdot [\mu_t U_\phi \mathbb{E}[\nabla f(x_0)]] \\
&= \nabla \cdot [U_\phi \nabla \mu_t + \mu_t U_\phi \mathbb{E}[\nabla f(x_0)]].
\end{aligned}$$

Combining our previous argument with the analysis of Vempala & Wibisono (2019), we have

$$\begin{aligned}
\partial_t \, \mathrm{D}_{\mathrm{KL}}(\mu_t \| \pi) &= \partial_t \int \mu_t(x) \log \frac{\mu_t(x)}{\pi(x)} dx \\
&= \int \partial_t \mu_t(x) \log \frac{\mu_t(x)}{\pi(x)} dx \\
&= \int \nabla \cdot [U_\phi \nabla \mu_t + \mu_t U_\phi \mathbb{E}[\nabla f(x_0)]] \log \frac{\mu_t(x)}{\pi(x)} dx \\
&= -\int \left\langle U_\phi \nabla \mu_t + \mu_t U_\phi \mathbb{E}[\nabla f(x_0)]], \nabla \log \frac{\mu_t(x)}{\pi(x)} \right\rangle dx \\
&= -\int \left\langle U_\phi \mu_t \nabla \log \mu_t + U_\phi \mu_t \nabla \log \pi - \mu_t \nabla \log \pi + \mu_t U_\phi \mathbb{E}[\nabla f(x_0)]], \nabla \log \frac{\mu_t(x)}{\pi(x)} \right\rangle dx \\
&= -\int \left\langle U_\phi \mu_t \nabla \log \frac{\mu_t}{\pi} + \mu_t U_\phi \mathbb{E}[\nabla f(x_0) - \nabla f(x_t)]], \nabla \log \frac{\mu_t(x)}{\pi(x)} \right\rangle dx \\
&= -\mathbb{E}[\|U_\phi^{1/2} \nabla \log \frac{\mu_t(x)}{\pi(x)}\|^2] + \mathbb{E}[\left\langle U_\phi^{1/2} \mathbb{E}[\nabla f(x_t) - \nabla f(x_0)], U_\phi^{1/2} \nabla \log \frac{\mu_t(x)}{\pi(x)} \right\rangle].
\end{aligned}$$

where we have used the fact that $U_\phi$ is a diagonal matrix with non-negative entries, so $U_\phi = U_\phi^{1/2} U_\phi^{1/2} = (U_\phi^{1/2})^T U_\phi^{1/2}$. Then we have (by Cauchy-Schwartz and Young's)

$$\begin{aligned}
\mathbb{E}[\left\langle U_\phi^{1/2} \mathbb{E}[\nabla f(x_t) - \nabla f(x_0)], U_\phi^{1/2} \nabla \log \frac{\mu_t(x)}{\pi(x)} \right\rangle] &\leq \mathbb{E}[\|U_\phi^{1/2} \mathbb{E}[\nabla f(x_t) - \nabla f(x_0)]\|^2] + \frac{1}{4}\mathbb{E}\|U_\phi^{1/2} \nabla \log \frac{\mu_t(x)}{\pi(x)}\|^2 \\
&= \mathbb{E}[\|U_\phi^{1/2}[\nabla f(x_t) - \nabla f(x_0)]\|^2] + \frac{1}{4}\mathbb{E}\|U_\phi^{1/2} \nabla \log \frac{\mu_t(x)}{\pi(x)}\|^2.
\end{aligned}$$

We can decompose the first term as

$$\mathbb{E}[\|U_\phi^{1/2}[\nabla f(x_t) - \nabla f(x_0)]\|^2] = \sum_{i=1}^{b} \phi_i \|U_k \nabla f(x_t) - U_k \nabla f(x_0)\|^2.$$

In line with the presentation in the draft Chewi (2024) we apply Lemma 16 from Chewi et al. (2021), which only requires smoothness and $L^2$ integrability in the marginal potential:

**Lemma 4** (Lemma 16 of Chewi et al. (2021)). *Assume probability measure $\pi \propto e^{-f(x)} \in \mathcal{P}^2(\mathbb{R}^d)$ has $L$-smooth potential $f$. Then for any probability measure $\mu$*

$$\mathbb{E}_\mu[\|\nabla f\|^2] \leq FI(\mu\|\pi) + 2dL.$$

By the smoothness of $f$, we have:

$$\begin{aligned}
\mathbb{E}\|U_k \nabla f(x_t) - U_k \nabla f(x_0)\|^2 &\leq 2L_i^2 \mathbb{E}\|x_t - x_0\|^2 = 2L_i^2 \mathbb{E}\|U_k t \nabla f(x_0) + U_k \sqrt{2} W_t\|^2 \\
&\leq 2L_i^2 t^2 \mathbb{E}\|U_k \nabla f(x_0) + U_k \nabla f(x_t) - U_k \nabla f(x_t)\|^2 + \mathbb{E}[2d_i L_i^2 t] \\
&\leq 2L_i^2 t^2 \mathbb{E}\|U_k \nabla f(x_0) - U_k \nabla f(x_t)\|^2 + 2L_i^2 \mathbb{E}\|U_k \nabla f(x_t)\|^2 + \mathbb{E}[2d_i L_i^2 t].
\end{aligned}$$

Suppose $t \leq \lambda_i \leq \frac{1}{2L_i}$, then

$$\mathbb{E}\|U_k \nabla f(x_t) - U_k \nabla f(x_0)\|^2 \leq \frac{1}{2}\mathbb{E}\|U_k \nabla f(x_0) - U_k \nabla f(x_t)\|^2 + 2L_i^2 \mathbb{E}\|U_k \nabla f(x_t)\|^2 + \mathbb{E}[2d_i L_i^2 t].$$

Hence

$$\mathbb{E}\|U_k \nabla f(x_t) - U_k \nabla f(x_0)\|^2 \leq 4L_i^2 \mathbb{E}\|U_k \nabla f(x_t)\|^2 + \mathbb{E}[4d_i L_i^2 t].$$

Plugging in Lemma 4 yields

$$\mathbb{E}\|U_k \nabla f(x_t) - U_k \nabla f(x_0)\|^2 \leq 4L_i^2 FI(\mu\|\pi) + \mathbb{E}[8t d_i L_i^3 + 4d_i L_i^2 t].$$

Assume $\lambda_i \leq \frac{\sqrt{\phi_{\min}}}{4L_i}$. Then

$$\begin{aligned}
\mathbb{E}[\sum_{i=1}^{b} \phi_i[4t^2 L_i^2 FI(\mu_{B_i}\|\pi_{B_i}) + 8dL_i^3 t^2 + 4d_i L_i^2 t]] &\leq \mathbb{E}[\sum_{i=1}^{b} \frac{\phi_{\min}\phi_i}{4} FI(\mu_{B_i}\|\pi_{B_i}) + \sum_{i=1}^{b} \phi_i[8dL_i^3 t^2 + 4d_i L_i^2 t]] \\
&\leq FI(\mu_t\|\pi)\mathbb{E}[\sum_{i=1}^{b} \phi_i \frac{\phi_{\min}}{4} + \sum_{i=1}^{b} \phi_i[8d_i L_i^3 t^2 + 4d_i L_i^2 t]] \\
&= \frac{\phi_{\min}}{4} FI(\mu_t\|\pi) + \mathbb{E}[\sum_{i=1}^{b} \phi_i[8d_i L_i^3 t^2 + 4d_i L_i^2 t]] \\
&\leq \frac{\phi_{\min}}{4} FI(\mu_t\|\pi) + \mathbb{E}[6d_i L_i^2 t].
\end{aligned}$$

We then have

$$\partial_t D_{KL}(\mu_t\|\pi) \leq -\frac{\phi_{\min}}{2} FI(\mu_t\|\pi) + 6\mathbb{E}[d_i L_i^2]t \leq -\phi_{\min}\gamma D_{KL}(\mu_t\|\pi) + 6\mathbb{E}[d_i L_i^2 t].$$

We start by multiplying both sides by $e^{-\phi_{\min}\gamma t}$ and integrating from $t = 0$ to $\lambda_i$

$$KL(\mu_{\lambda_i}\|\pi) \leq e^{-\phi_{\min}\lambda_i} D_{KL}(\mu_0\|\pi) + 3\mathbb{E}[d_i L_i^2 \lambda_i^2].$$

Taking expectation over $i$ then gives the result

$$\mathbb{E}_i[KL(\mu_{\lambda_i}\|\pi)] \leq \mathbb{E}_i[e^{-\gamma\phi_{\min}\lambda_i}] D_{KL}(\mu_0\|\pi) + 3\mathbb{E}[d_i L_i^2 \lambda_i^2].$$

Iterating C.4 gives

$$\mathbb{E}[KL(\mu^k\|\pi)] \leq \mathbb{E}[e^{-\gamma\phi_{\min}\lambda_{\min}}]^k \, \mathrm{D}_{\mathrm{KL}}(\mu_0\|\pi) + 3\mathbb{E}[d_i L_i^2 \lambda_i^2] \sum_{i=0}^{k} \mathbb{E}[e^{-\gamma\phi_{\min}\lambda_{\min}}]^i$$

$$\leq e^{-\gamma\phi_{\min}\lambda_i k} \, \mathrm{D}_{\mathrm{KL}}(\mu_0\|\pi) + \frac{4}{\gamma\phi_{\min}\lambda_{\min}} \mathbb{E}[d_i L_i^2 \lambda_i^2],$$

where we first bound using the minimum step size, then apply the power series bound

$$\sum_{i=0}^{k} \mathbb{E}[e^{-\gamma\phi_{\min}\lambda_i}]^i \leq \sum_{i=0}^{k} e^{-\gamma\phi_{\min}\lambda_{\min}} \leq \frac{1}{1 - e^{-\gamma\phi_{\min}\lambda_{\min}}}$$

and then apply $\frac{1}{1-e^{-a}} \leq \frac{4}{3a}$ to obtain

$$\frac{1}{1 - e^{-\gamma\phi_{\min}\lambda_{\min}}} \leq \frac{4}{3\gamma\phi_{\min}\lambda_{\min}}.$$

## D   CYCLIC BLOCK LANGEVIN DIFFUSION

In this section we provide proofs relating to Cyclic Block Langevin Diffusion (CBLD, the focus of the main text) and a time-discretized version, Cyclic Block Langevin Monte Carlo (CBLMC).

The CBLD sampling algorithm is shown in Algorithm 4:

---
**Algorithm 4** Cyclic Block Langevin Diffusion (CBLD)

---
1: **procedure** CBLD($x_0 \in \mathrm{dom}(f)$, Block Permutation $\sigma = \{B_1, ..., B_b\}$, Step Sizes $\lambda \in \mathbb{R}_+^b$)
2:     **for** $k \geq 0$ **do**
3:         Define $\tau_0 = t_{kb}$
4:         **for** $n = 1$ to $b$ **do**
5:             Choose $i = \sigma_n$ and set $\tau_n = \tau_{n-1} + \lambda_i$
6:             Sample:

$$x_{\tau_n} = x_{\tau_{n-1}} - \int_{\tau_{n-1}}^{\tau_n} U_i \nabla f(x) dt + \int_{\tau_{n-1}}^{\tau_n} U_i \sqrt{2\beta^{-1}\lambda_i} dW_t$$

7:         **end for**
8:         Set: $x_{t_{(k+1)b}} = x_{\tau_b}$
9:             $t_{(k+1)b} = \tau_b$
10:     **end for**
11: **end procedure**

---

A crucial identity used in our analysis is the "chain lemma" for KL-divergence. For any two distributions $\mu$

$$\mathrm{D}_{\mathrm{KL}}(\mu_t\|\pi) = \mathbb{E}[\mathrm{D}_{\mathrm{KL}}(\mu_{t|B}\|\pi_{|B})] + \mathrm{D}_{\mathrm{KL}}(\mu_{t,B}|\pi_B)$$

where $B$ is a subspace of $\mathbb{R}^d$, $\mathrm{D}_{\mathrm{KL}}(\mu_{t|B}\|\pi_{|B})$ is the KL-divergence of $\mu_t$ and $\pi$ conditioned on an element of $B$, and $\mathrm{D}_{\mathrm{KL}}(\mu_{t,B}|\pi_B)$ is the KL-divergence of $\mu_t$ and $\pi$ marginalized over $\mathbb{R} \setminus B$. We also state two trivial lemmas for any $\gamma$-LSI distribution $\nu$. We first state an equivalent definition of Assumption 2.

**Definition 1** (Alternative LSI). $\pi \propto \exp[-\beta f(x)]$ satisfies a log-Sobolev inequality (LSI) with $C_{LSI} = \frac{1}{\gamma}$ if for all smooth $g$:

$$\mathbb{E}_\pi[g^2 \log g^2] - \mathbb{E}_\pi[g^2] \log \mathbb{E}_\pi[g^2] \leq \frac{1}{2\gamma} \mathbb{E}_\pi[\|\nabla g\|^2]$$

where the equivalence with the previous statement follows by choosing $g^2(x) = \frac{\mu(x)}{\pi(x)}$.

**Lemma 5.** *Suppose $A, B$ are disjoint subspaces of $\mathbb{R}^d$ with $A \cup B = \mathbb{R}^d$. Then the $A$ marginal $\nu_A$ also satisfies $\gamma$-LSI.*

*Proof.* By the LSI, for any smooth $g : \mathbb{R}^d \to \mathbb{R}$
$$\mathbb{E}_\nu \left[ g^2 \log g^2 \right] - \mathbb{E}_\nu \left[ g^2 \right] \log \mathbb{E}_\nu \left[ g^2 \right] \le \mathbb{E}_\nu \left[ \|\nabla g\|^2 \right].$$
For $g : A \to \mathbb{R}$, we can re-express the terms as
$$\mathbb{E}_{\nu_{B|A}} \mathbb{E}_{\nu_A} \big[ \left[ g^2 \log g^2 \right] - \mathbb{E}_{\nu_{B|A}} \left( \mathbb{E}_{\nu_A} \left[ g^2 \right] \log \mathbb{E}_\nu \left[ g^2 \right] \right) \le \mathbb{E}_{\nu_{B|A}} \mathbb{E}_{\nu_A} \left[ \|\nabla g\|^2 \right].$$
Since $\mathbb{E}_{\nu_{B|A}}[g(z)] = g(z)$ for all $z \in A$, we simplify to
$$\mathbb{E}_{\nu_A} \left[ g^2 \log g^2 \right] - \mathbb{E}_{\nu_A} \left[ g^2 \right] \log \mathbb{E}_\nu \left[ g^2 \right] \le \mathbb{E}_{\nu_A} \left[ \|\nabla g\|^2 \right].$$
$\square$

Recall that the sub-step dynamics are described by the SDE
$$dx = U_n \left[ -\nabla f(x) dt + \sqrt{2\beta} dW_t^n \right]. \tag{10}$$
where $dW_t^n$ denotes $d_n$-dimensional Brownian noise. We can then derive the coordinate Fokker-Planck equation:

**Lemma 6.** *Let $\mu_{t|x_0}$ be the law of $x$ at time $t \in [0, \lambda_n]$ described by the SDE in Equation (10), where $\mu_{t|x_0}$ is conditioned on the starting state $x_0$. Then $\partial_t \mu_{t, \overline{B}_n | x_0} = 0$ and*
$$\partial_t \mu_{t, B_n | \overline{B}_n, x_0} = \beta^{-1} \nabla^2 \cdot \mu_{B_n | \overline{B}_n, x_0} + \nabla \cdot (\mu_{B_n | \overline{B}_n, x_0} \nabla f(x_t))$$
*is the Fokker-Planck equation for the subspace diffusion.*

*Proof.* The second claim is trivially shown using Itô's Lemma. Note that since $\mu_{t, B_n | \overline{B}_n, x_0}$ is only supported on $B_n$:

1. $\text{Tr}[\beta^{-1} U_n \nabla^2 \mu_{t, B_n | \overline{B}_n, x_0}] = \beta^{-1} \nabla^2 \cdot \mu_{t, B_n | \overline{B}_n, x_0}$

2. $\nabla \cdot \mu_{t, B_n | \overline{B}_n, x_0} \nabla f(x_t) = \nabla \cdot \mu_{t, B_n | \overline{B}_n, x_0} \nabla f(x_t)$

Then we have
$$\partial_t \mu_{t, B_n | \overline{B}_n, x_0} = \beta^{-1} \nabla^2 \cdot \mu_{B_n | \overline{B}_n, x_0} + \nabla \cdot (\mu_{B_n | \overline{B}_n, x_0} \nabla f(x_t)).$$
We now use this to prove the first claim.

Consider the law of $x$ in sub-step $n$ conditioned on the initial state $x_0$ given by $\mu_{t|x_0}$. Note that $\mu_{t|x_0} = \mu_{t, B_n | \overline{B}_n, x_0} \mu_{t, \overline{B}_n | x_0}$.

By the Fokker-Planck equation associated with the SDE and the product rule, we have:
$$\partial_t \mu_{t|x_0} = \beta^{-1} \text{Tr}[U_n^T \nabla^2 \mu_{t|x_0}] + \nabla \cdot (\mu_{t|x_0} U_n \nabla f(x))$$
$$\mu_{t, \overline{B}_n | x_0} \partial_t \mu_{t, B_n | \overline{B}_n, x_0} + \mu_{t, B_n | \overline{B}_n, x_0} \partial_t \mu_{t, \overline{B}_n | x_0} = \beta^{-1} \text{Tr}[U_n^T \nabla^2 \mu_{t, B_n | \overline{B}_n, x_0} \mu_{t, \overline{B}_n | x_0}]$$
$$+ \nabla \cdot (\mu_{t, B_n | \overline{B}_n, x_0} \mu_{t, \overline{B}_n | x_0} U_n \nabla f(x)).$$
Note that
$$\beta^{-1} \text{Tr}[U_n^T \nabla^2 \mu_{t, B_n | \overline{B}_n, x_0} \mu_{t, \overline{B}_n | x_0}] = \beta^{-1} \mu_{t, \overline{B}_n | x_0} \nabla^2 \cdot \mu_{t, B_n | \overline{B}_n, x_0}$$
$$\text{and}$$
$$\nabla \cdot (\mu_{t, B_n | \overline{B}_n, x_0} \mu_{t, \overline{B}_n | x_0} U_n \nabla f(x)) = \mu_{t, \overline{B}_n | x_0} \nabla \cdot (\mu_{t, B_n | \overline{B}_n, x_0} \nabla f(x)).$$
We then have
$$\mu_{t, \overline{B}_n | x_0} \overbrace{(\partial_t \mu_{t, B_n | \overline{B}_n, x_0} - \beta^{-1} \nabla^2 \cdot \mu_{t, B_n | \overline{B}_n, x_0} - \nabla \cdot (\mu_{t, B_n | \overline{B}_n, x_0} \nabla f(x)))}^{①} = \mu_{t, B_n | \overline{B}_n, x_0} \partial_t \mu_{t, \overline{B}_n | x_0}.$$

We assume that $\mu_t$ is supported on $\mathbb{R}^d$, therefore $\mu_{t|x_0} = \mu_{t, \overline{B}_n | x_0} \mu_{t, B_n | \overline{B}_n, x_0} > 0$.

As previously discussed, Itô's lemma implies ① is 0. For equality to hold, then, $\partial_t \mu_{t, \overline{B}_n} = 0$. $\square$

We prove the following technical lemma for later use in the descent bound:

**Lemma 7.** *Suppose $A, B$ are disjoint subspaces of $\mathbb{R}^d$. Then we have*

$$\mathrm{D_{KL}}(\mu_A \| \pi_A) \leq \mathrm{D_{KL}}(\mu_{A|B} \| \pi_{A|B}).$$

*Proof.* Note that for all $x \in A$

$$\mu_A(x) = \int_B \mu_{A,B}(x, y) dy = \int_B \mu_A(x|y) \mu_B(y) dy = \mathbb{E}_{y \in B}[\mu_A(x|y)] \triangleq \mathbb{E}_B[\mu_{A|B}].$$

By the convexity of the KL-divergence and Jensen's Inequality

$$\mathrm{D_{KL}}(\mu_A \| \pi_A) = KL(\mathbb{E}_B[\mu_{A|B}] \| \mathbb{E}_B[\pi_{A|B}]) \leq \mathbb{E}_B[\mathrm{D_{KL}}(\mu_{A|B} \| \pi_{A|B})] \triangleq \mathrm{D_{KL}}(\mu_{A|B} \| \pi_{A|B}).$$

$\square$

Lemma 7 can be considered a restatement of the "data processing inequality". Removing the conditioning on subspace $B$ effectively reduces the available information, akin to a noisy channel, decreasing the divergence between distributions.

**Lemma 8.**

$$\mathrm{D_{KL}}(\mu_n \| \pi) \leq e^{-2\gamma\beta^{-1}\lambda_n} \left[\mathrm{D_{KL}}(\mu_{n-1} \| \pi)\right] + (1 - e^{-2\gamma\beta^{-1}\lambda_n}) \mathrm{D_{KL}}(\mu_{n-1,\overline{B}_1} \| \pi_{\overline{B}_1})$$

*Proof.* Using Lemma 6, we can show by standard arguments Vempala & Wibisono (2019); Chewi et al. (2021) that within sub-step $n$:

$$\mathrm{D_{KL}}(\mu_{t,\overline{B}_1} \| \pi_{\overline{B}_1}) \leq e^{-2\gamma\beta^{-1}t} \mathrm{D_{KL}}(\mu_{0,\overline{B}_1} \| \pi_{\overline{B}_1}) \tag{11}$$

Using (11) and the chain rule for KL-divergence

$$\begin{aligned}
\mathrm{D_{KL}}(\mu_n \| \pi) &= \mathbb{E}\left[\mathrm{D_{KL}}(\mu_{n,B_1|\overline{B}_1} \| \pi_{B_1|\overline{B}_1})\right] + \mathrm{D_{KL}}(\mu_{n-1,\overline{B}_1} \| \pi_{\overline{B}_1}) \\
&\leq e^{-2\gamma\lambda_n\beta^{-1}} \mathbb{E}\left[\mathrm{D_{KL}}(\mu_{n-1,B_1|\overline{B}_1} \| \pi_{B_1|\overline{B}_1})\right] + \mathrm{D_{KL}}(\mu_{n-1,\overline{B}_1} \| \pi_{\overline{B}_1}) \\
&= e^{-2\gamma\lambda_n\beta^{-1}} \left[\mathrm{D_{KL}}(\mu_{n-1,B_1|\overline{B}_1} \| \pi_{B_1|\overline{B}_1}) - \mathrm{D_{KL}}(\mu_{n-1,\overline{B}_1} \| \pi_{\overline{B}_1}))\right] + \mathrm{D_{KL}}(\mu_{n-1,\overline{B}_1} \| \pi_{\overline{B}_1}) \\
&= e^{-2\gamma\lambda_n\beta^{-1}} \mathrm{D_{KL}}(\mu_{n-1} \| \pi) + (1 - e^{-2\gamma\lambda_n\beta^{-1}}) \mathrm{D_{KL}}(\mu_{n-1,\overline{B}_1} \| \pi_{\overline{B}_1}).
\end{aligned}$$

$\square$

An immediate consequence of Lemma 8 is that the KL-divergence is non-increasing, as stated in the following Corollary.

**Corollary 1.** *For all $i \in \{1, ..., b\}$, $\mathrm{D_{KL}}(\mu_i \| \pi) \leq \mathrm{D_{KL}}(\mu_0 \| \pi)$*

### D.1 PROOF OF LEMMA 1

*Proof.* We prove the claim by induction on $b$. The claim is immediately evident for $b = 1$ as a consequence of 8 with $C_{\max} = C_1$, since $\overline{B}_1 = \emptyset$.

Now we assume the inductive hypothesis for some $b - 1 \geq 1$ and prove the claim for $b \geq 2$ blocks.

We start by applying Lemma 8 twice to obtain terms relating to step $b - 2$, obtaining

$$\mathrm{D_{KL}}(\mu_b \| \pi) \leq C_b \mathrm{D_{KL}}(\mu_{b-1} \| \pi) + (1 - C_b) \mathrm{D_{KL}}(\mu_{b-1,\overline{B}_b} \| \pi_{\overline{B}_b}) + D_b$$

$$\text{Second descent expansion: } \leq C_b C_{b-1} \mathrm{D_{KL}}(\mu_{b-2} \| \pi) + (1 - C_b) \mathrm{D_{KL}}(\mu_{b-1,\overline{B}_b} \| \pi_{\overline{B}_b})$$

$$+ C_b(1 - C_{b-1}) \mathrm{D_{KL}}(\mu_{b-2,\overline{B}_{b-1}} \| \pi_{\overline{B}_{b-1}}) + D_b + C_b D_{b-1}$$

.

From here, we note that $\mathrm{D_{KL}}(\mu_{b-1,\overline{B}_b}\|\pi_{\overline{B}_b})$ satisfies the theorem conditions, since all blocks in $\overline{B}_b$ have been sampled. We can therefore apply the inductive hypothesis and obtain

$$\mathrm{D_{KL}}(\mu_b\|\pi) \leq C_b C_{b-1} \mathrm{D_{KL}}(\mu_{b-2}\|\pi) + C_{\max}(1-C_b)\mathrm{D_{KL}}(\mu_{0,\overline{B}_b}\|\pi_{\overline{B}_b}) + (1-C_b)\sum_{i=1}^{b-1} D_i$$
$$+ C_b(1-C_{b-1})\mathrm{D_{KL}}(\mu_{b-2,\overline{B}_{b-1}}\|\pi_{\overline{B}_{b-1}}) + D_b + C_b D_{b-1}.$$

Using Lemma 7, we can upper bound

$$\mathrm{D_{KL}}(\mu_{b-2,\overline{B}_{b-1}}\|\pi_{\overline{B}_{b-1}}) \leq \mathrm{D_{KL}}(\mu_{b-2,\overline{B}_{b-1}|B_{b-1}}\|\pi_{\overline{B}_{b-1}|B_{b-1}})$$

and then apply the chain lemma

$$\mathrm{D_{KL}}(\mu_{b-2,\overline{B}_{b-1}|B_{b-1}}\|\pi_{\overline{B}_{b-1}|B_{b-1}}) = \mathrm{D_{KL}}(\mu_{b-2}\|\pi) - \mathrm{D_{KL}}(\mu_{b-2,B_{b-1}}\|\pi_{B_{b-1}})$$

to obtain

$$\mathrm{D_{KL}}(\mu_b\|\pi) \leq C_b C_{b-1} \mathrm{D_{KL}}(\mu_{b-2}\|\pi)$$
$$+ C_b(1-C_{b-1})\mathrm{D_{KL}}(\mu_{b-2}\|\pi) - C_b(1-C_{b-1})\mathrm{D_{KL}}(\mu_{b-2,B_{b-1}}\|\pi_{B_{b-1}})$$
$$+ (1-C_b)\sum_{i=1}^{b-1} D_i + C_{\max}(1-C_b)\mathrm{D_{KL}}(\mu_{0,\overline{B}_b}\|\pi_{\overline{B}_b}) + D_b + C_b D_{b-1}.$$

We can define $\overline{B}_{b,b-1} \triangleq \overline{B}_b \cap \overline{B}_{b-1}$ (all variable blocks except the last two) and apply the chain lemma

$$\mathrm{D_{KL}}(\mu_{0,\overline{B}_b}\|\pi_{\overline{B}_b}) = \mathrm{D_{KL}}(\mu_{0,\overline{B}_{b,b-1}|B_{b-1}}\|\pi_{\overline{B}_{b,b-1}|B_{b-1}}) + \mathrm{D_{KL}}(\mu_{0,B_{b-1}}\|\pi_{B_{b-1}}).$$

to obtain the bound

$$\mathrm{D_{KL}}(\mu_b\|\pi) \leq C_b C_{b-1} \mathrm{D_{KL}}(\mu_{b-2}\|\pi)$$
$$+ C_b(1-C_{b-1})\mathrm{D_{KL}}(\mu_{b-2}\|\pi) - C_b(1-C_{b-1})\mathrm{D_{KL}}(\mu_{b-2,B_{b-1}}\|\pi_{B_{b-1}})$$
$$+ (1-C_b)\sum_{i=1}^{b-1} D_i$$
$$+ C_{\max}(1-C_b)\mathrm{D_{KL}}(\mu_{0,\overline{B}_{b,b-1}|B_{b-1}}\|\pi_{\overline{B}_{b,b-1}|B_{b-1}}) + C_{\max}(1-C_b)\mathrm{D_{KL}}(\mu_{0,B_{b-1}}\|\pi_{B_{b-1}})$$
$$+ D_b + C_b D_{b-1}.$$

We can regroup the terms and cancel $C_B D_{b-1} - C_B D_{b-1} = 0$ yields

$$\mathrm{D_{KL}}(\mu_b\|\pi) \leq C_b \mathrm{D_{KL}}(\mu_{b-2}\|\pi)$$

$$- C_b\left(C_{\max}\mathrm{D_{KL}}(\mu_{0,\overline{B}_{b,b-1}|B_{b-1}}\|\pi_{\overline{B}_{b,b-1}|B_{b-1}}) + \sum_{i=1}^{b-2} D_i\right)$$
$$- C_b(1-C_{b-1})\mathrm{D_{KL}}(\mu_{b-2,B_{b-1}}\|\pi_{B_{b-1}})$$
$$+ C_{\max}\mathrm{D_{KL}}(\mu_{0,\overline{B}_{b,b-1}|B_{b-1}}\|\pi_{\overline{B}_{b,b-1}|B_{b-1}}) + C_{\max}(1-C_b)\mathrm{D_{KL}}(\mu_{0,B_{b-1}}\|\pi_{B_{b-1}})$$
$$+ D_b + \sum_{i=1}^{b-1} D_i.$$

By applying the inductive hypothesis in reverse, we can show

$$-C_b\left(C_{\max}\mathrm{D_{KL}}(\mu_{0,\overline{B}_{b-1}|B_{b-1}}\|\pi_{\overline{B}_{b,b-1}|B_{b-1}}) + \sum_{i=1}^{b,b-1} D_i\right) \leq -C_b\mathrm{D_{KL}}(\mu_{b-1,\overline{B}_{b,b-1}|B_{b-1}}\|\pi_{\overline{B}_{b,b-1}|B_{b-1}}).$$

Substituting this into the second line, we have

$$\mathrm{D_{KL}}(\mu_b\|\pi) \leq C_b \mathrm{D_{KL}}(\mu_{b-2}\|\pi)$$
$$- C_b\mathrm{D_{KL}}(\mu_{b-2,\overline{B}_{b,b-1}|B_{b-1}}\|\pi_{\overline{B}_{b,b-1}|B_{b-1}}) - C_b(1-C_{b-1})\mathrm{D_{KL}}(\mu_{b-2,B_{b-1}}\|\pi_{B_{b-1}})$$
$$+ C_{\max}\mathrm{D_{KL}}(\mu_{0,\overline{B}_{b,b-1}|B_{b-1}}\|\pi_{\overline{B}_{b,b-1}|B_{b-1}}) + C_{\max}(1-C_b)\mathrm{D_{KL}}(\mu_{0,B_{b-1}}\|\pi_{B_{b-1}})$$
$$+ \sum_{i=1}^{b} D_i.$$

We can once again expand the terms

$$D_{KL}(\mu_{b-2,\overline{B}_{b,b-1|B_{b-1}}} \| \pi_{\overline{B}_{b,b-1|B_{b-1}}})$$

$$C_{\max} D_{KL}(\mu_{0,\overline{B}_{b,b-1|B_{b-1}}} \| \pi_{\overline{B}_{b,b-1|B_{b-1}}}).$$

Using the chain lemma and canceling the single block terms gives

$$
\begin{aligned}
D_{KL}(\mu_b \| \pi) \leq & C_b D_{KL}(\mu_{b-2} \| \pi) \\
& - C_b D_{KL}(\mu_{b-2,\overline{B}_b} \| \pi_{\overline{B}_b}) + C_b C_{b-1} D_{KL}(\mu_{b-2,B_{b-1}} \| \pi_{B_{b-1}}) \\
& + C_{\max} D_{KL}(\mu_{0,\overline{B}_b} \| \pi_{\overline{B}_b}) - C_b C_{\max} D_{KL}(\mu_{0,B_{b-1}} \| \pi_{B_{b-1}}) \\
& + \sum_{i=1}^{b} D_i.
\end{aligned}
$$

Since $C_{\max} \geq C_{b-1}$ by definition and $D_{KL}$ is non-increasing with respect to inner loop steps $b$, we can disregard

$$
\begin{aligned}
& C_b C_{b-1} D_{KL}(\mu_{b-2,B_{b-1}} \| \pi_{B_{b-1}}) - C_b C_{\max} D_{KL}(\mu_{0,B_{b-1}} \| \pi_{B_{b-1}}) \\
& \leq C_b C_{b-1} D_{KL}(\mu_{b-2,B_{b-1}} \| \pi_{B_{b-1}}) - C_b C_{\max} D_{KL}(\mu_{b-2,B_{b-1}} \| \pi_{B_{b-1}}) \leq 0.
\end{aligned}
$$

We now add zero to the right hand side via

$$0 = C_b D_{KL}(\mu_{b-2,B_b|\overline{B}_b} \| \pi_{B_b|\overline{B}_b}) - C_b D_{KL}(\mu_{b-2,B_b|\overline{B}_b} \| \pi_{B_b|\overline{B}_b}).$$

We then have the three terms

$$C_b D_{KL}(\mu_{b-2} \| \pi) + \sum_{i=1}^{b} D_i,$$

$$-C_b D_{KL}(\mu_{b-2,\overline{B}_b} \| \pi_{\overline{B}_b}) - C_b D_{KL}(\mu_{b-2,B_b|\overline{B}_b} \| \pi_{B_b|\overline{B}_b}), \tag{12}$$

$$C_{\max} D_{KL}(\mu_{0,\overline{B}_b} \| \pi_{\overline{B}_b}) + C_b D_{KL}(\mu_{b-2,B_b|\overline{B}_b} \| \pi_{B_b|\overline{B}_b}). \tag{13}$$

Note that the previous time steps left $\mu_{b-2,B_b|\overline{B}_b}$ invariant, hence $D_{KL}(\mu_{b-2,B_b|\overline{B}_b} \| \pi_{B_b|\overline{B}_b}) = D_{KL}(\mu_{0,B_b|\overline{B}_b} \| \pi_{B_b|\overline{B}_b})$. Then by applying the chain lemma to (13) and (12), we obtain

$$
\begin{aligned}
D_{KL}(\mu_b \| \pi) \leq & C_b D_{KL}(\mu_{b-2} \| \pi) - C_b D_{KL}(\mu_{b-2} \| \pi) + C_{\max} D_{KL}(\mu_0 \| \pi) + \sum_{i=1}^{b} D_i \\
= & C_{\max} D_{KL}(\mu_0 \| \pi) + \sum_{i=1}^{b} D_i
\end{aligned}
$$

which completes the proof. $\qquad \square$

### D.2  CBLMC: EULER-MARUYAMA DISCRETIZATION

As discussed in the main text, Lemma 1 can be used to trivially bound both the continuous ($C_i = e^{-2\gamma\lambda_i\beta^{-1}}$, $D_i = 0$) and discrete ($C_i = e^{-\gamma\lambda_i\beta^{-1}}$, $D_i = 3d_i L_i^2 \lambda_i^2$) cases. As with RBLMC in the previous section, we take $\beta = 1$ to simplify constant terms. Discrete-time CBLMC is shown in Algorithm 5

We recall the convergence results for LMC from Vempala & Wibisono (2019) (adjusted using Lemma 16 of Chewi et al. (2021) as in our analysis of RBLMC).

**Theorem 4** (LD Convergence (Theorem 2 of Vempala & Wibisono (2019))). *Let $\pi$ be a distribution satisfying an LSI with constant $\gamma$ with $L$-smooth potential. Assume that the LMC step size $\lambda$ is chosen such that $\lambda \in \left(0, \frac{\gamma}{4L^2}\right]$. Then after a single step of LMC, the distribution $\mu_{k+1}$ satisfies*

$$D_{KL}(\mu_{k+1} \| \pi) \leq e^{-\gamma\lambda} D_{KL}(\mu_k \| \pi) + 3L^2 d\lambda^2$$

---

**Algorithm 5** Cyclic Block Langevin Monte Carlo (CBLMC)

---

1: **procedure** CBLD($x_0 \in \text{dom}(f)$, Block Permutation $\sigma = \{B_1, ..., B_b\}$, Step Sizes $\lambda \in \mathbb{R}_+^b$)
2:     **for** $k \geq 0$ **do**
3:         Set $x_0^{k+1} = x^k$
4:         **for** $n = 1$ to $b$ **do**
5:             Choose $i = \sigma_n$, sample $\xi_k \sim \mathcal{N}(0, I^d)$

$$x_{n+1}^{k+1} = x_n^{k+1} - \lambda_i U_i \nabla f(x_n^{k+1}) + U_i \sqrt{2\lambda_i} \xi_k$$

6:         **end for**
7:         Set $x^{k+1} = x_{b+1}^{k+1}$
8:     **end for**
9: **end procedure**

---

For CBLMC, we additionally assume that the potential is $L$-smooth (Assumption 4). From Beck & Tetruashvili (2013), this implies each block has a separate smoothness constant $L_i \leq L$. From applying Theorem 4, each block step has the descent

$$\text{D}_{\text{KL}}(\mu^{kb}\|\pi) \leq e^{-\gamma\lambda_i} \text{D}_{\text{KL}}(\mu^{b-1}\|\pi) + (1 - e^{-\gamma\lambda_i}) \text{D}_{\text{KL}}(\mu_{B_i}^{b-1}\|\pi_{\overline{B_i}}) + 3L_i^2 d_i \lambda_i^2.$$

When iterated for $kb$ cycles, we obtain the bound

$$\text{D}_{\text{KL}}(\mu^{kb}\|\pi) \leq e^{-\gamma kb\lambda_{\min}kb} \text{D}_{\text{KL}}(\mu^0\|\pi) + \frac{4}{\gamma\lambda_{\min}} \sum_{i=1} L_i^2 d_i \lambda_i^2.$$

where the constant terms are derived analogously to the RBLMC proof in the preceding section.

## E    PROOF OF THEOREM 3

We begin by recalling the following Lemmas from literature:

**Lemma 9** (Uniform $L^2$ bound on Langevin Diffusion (Lemma 3 of Raginsky et al. (2017))). *Let* $f : \mathbb{R}^d \to \mathbb{R}$ *be a differentiable function satisfying Assumption 5. For a random variable* $x(t) = x(0) - \int_0^t \nabla f(x(s))ds + \int_0^t dW_s$, *we have the bound*

$$\mathbb{E}[\|x(t)\|^2] \leq \mathbb{E}[\|x(0)\|^2]e^{-mt} + \frac{d/\beta + c}{m}(1 - e^{-2mt}).$$

**Lemma 10** (Wasserstein bound from Relative Entropy (Corollary 2.3 of Bolley & Villani (2005))). *Let* $\mu$, $\nu$ *be two probability measures on some measurable space* $X$ *equipped with measurable distance* $\mathscr{D}$, *and let* $\phi : X \to \mathbb{R}^+$ *be a non-negative measurable function. Assume that* $\exists x_0 \in X$, $\alpha > 0$ *such that* $\int e^{\alpha \mathscr{D}(x_0,x)^p} d\nu(x)$ *is finite. Then*

$$W_2 \leq C \left[ \text{D}_{\text{KL}}(\mu\|\nu)^{1/2} + \left( \frac{\text{D}_{\text{KL}}(\mu\|\nu)}{2} \right)^{1/4} \right]$$

*where*

$$C \triangleq 2 \inf_{x_0 \in X} \left( \frac{1}{\alpha}(\frac{3}{2} + \log \int e^{\alpha \mathscr{D}(x_0,x)^p} d\nu(x)) \right)^{1/p}.$$

In addition, we adapt the following Lemma from Raginsky et al. (2017)

**Lemma 11** (Exponential $L^2$ Integrability of Block Langevin Diffusion). *Let* $f : \mathbb{R}^d \to \mathbb{R}$ *be a differentiable function satisfying Assumption 5, and let* $x^k(t) = x(0) - \int_0^t U_k \nabla f(x(s))ds + \int_0^t U_k dW_s$ *be a random variable in* $\mathbb{R}^d$ *across some number of iterations* $k$, *where* $\sum_{i=1}^b U_i = I_d$. *Suppose the initial state* $x_0$ *is drawn from some* $\mu_0$ *satisfying Assumption 6 and* $\beta > 2/m$. *Then on iteration* $k$

$$\log E\left[e^{\|x_\lambda^k\|^2}\right] \leq \kappa_0 + 2(c + \frac{d_{\max}}{\beta})k\lambda.$$

*Proof.* Define $G(x_t^k) \triangleq e^{\|x_t^k\|^2}$. By Itô's lemma, on iteration $k$ of $BLD$ we have

$$dG(x_t^k) = -2\left\langle x_t^k, U_k \nabla f(x_t^k) \right\rangle e^{\|x_t^k\|^2} dt + \frac{2\beta^{-1}}{2} \mathrm{Tr}\left[ U_k^2 (2e^{\|x_t^k\|^2} I + 4xx^T e^{\|x_t^k\|^2}) \right] dt$$

$$+ 2\sqrt{2\beta} \left\langle x_t^k, U_k \right\rangle e^{\|x_t^k\|^2} dW_t$$

$$= -2\left\langle x_t^k, U_k \nabla f(x_t^k) \right\rangle G(t) dt$$

$$+ 2d_k \beta^{-1} G(x_t^k) dt + 4\|U_k x_t^k\|^2 \beta^{-1} G(x_t^k) dt + 2\sqrt{2\beta} \left\langle x_t^k, U_k \right\rangle G(x_t^k) dW_t.$$

Integrating and summing across $k$ steps, we obtain

$$G(x_\lambda^k) = G(x^0) + 2\sum_{i=1}^{k} \left[ \int_0^\lambda \left[ -\left\langle x_t^k, U_k \nabla f(x_t^k) \right\rangle + \beta^{-1}\|U_k x_t^k\|^2 \right] G(x_t^k) dt \right.$$

$$\left. + \int_0^\lambda d_k \beta^{-1} G(x_t^k) dt + \int_0^\lambda \sqrt{2\beta} \left\langle x_t^k, U_k \right\rangle G(x_t^k) dW_t \right].$$

Applying the dissapativity condition and assuming $\beta > 2/m$, we can bound the first integrand as

$$-\left\langle x_t^k, U_k \nabla f(x_t^k) \right\rangle + 2\beta^{-1}\|U_k x_t^k\|^2 \le (2\beta^{-1} - m) \sum_{j \in B_i} (x_{t,j}^k)^2 + c \le c$$

which results in

$$G(x_\lambda^k) = G(x^0) + \sum_{i=1}^{k} 2(c + d_k \beta^{-1}) \int_0^\lambda G(x_t^k) dt + \int_0^\lambda 2\sqrt{2\beta} G(x_t^k) \left\langle x_t^k, U_k dW_t \right\rangle.$$

As stated in Raginsky et al. (2017), each Itô integral $\int_0^\lambda 2\sqrt{2\beta} G(x_t^k) \left\langle x_t^k, U_k dW_t \right\rangle$ is a zero-mean Martingale. Taking expectations over both sides and applying Assumption 6 yields

$$\mathbb{E}[G(x_\lambda^k)] = \mathbb{E}[G(x^0)] + \sum_{i=1}^{k} 2(c + d_k \beta^{-1}) \int_0^\lambda \mathbb{E}[G(x_t^k)] dt$$

$$\le e^{\kappa_0} + 2(c + d_{\max}\beta^{-1}) \int_0^{k\lambda} \mathbb{E}[G(x_t^k)] dt.$$

where the integrability of $\mathbb{E}[G(x_t^k)]$ across block steps follows from the continuity of $x_t^k$ across each block step $k$. By Grönwall's inequality, we have

$$\mathbb{E}[G(x_\lambda^k)] \le e^{\kappa_0} e^{2(c + d_{\max}\beta^{-1})k\lambda}$$

$$\log \mathbb{E}[G(x_\lambda^k)] \le \kappa_0 + 2(c + d_{\max}\beta^{-1})k\lambda$$

$\square$

Theorem 3 follows as a consequence of Lemma 2 by applying the Otto-Villani theorem coupled with the triangle inequality for $W_2$ as stated in the main text.

### E.1 PROOF OF LEMMA 2

*Proof.* Let $\mu_t^k$ and $\nu_t^k$ be the laws of SGBLD and BLD at times $t$ and iteration $k$ respectively with iterates $x^k(s)$, $y^k(s)$. We assume that each process selects the same variable blocks at each iteration, i.e. $B_x^k = B_y^k$.

Using the Girsanov formula, we can express the Radon-Nikodym derivative $\frac{d\nu_t^k}{d\mu_t^k}$ as

$$\frac{d\nu_t^k}{d\mu_t^k} = \exp\left[ \frac{\beta}{2} \int_0^t \left\langle U_k \nabla f(y^k(s)) - U_k g_z(y^k(t)), -U_k \nabla f(y^k(s)) ds + \sqrt{2\beta^{-1}} U_k dW_s \right\rangle \right.$$

$$\left. + \frac{\beta}{4} \int_0^t \left\langle U_k \nabla f(y^k(s)) - U_k g_z(y^k(s)), U_k \nabla f(y^k(s)) + U_k g_z(y^k(s)) \right\rangle dt \right]$$

$$= \exp\left[ \sqrt{\frac{\beta}{2}} \int_0^t \left\langle U_k \nabla f(y^k(s)) - U_k g_z(y^k(s)), dW_s \right\rangle - \frac{\beta}{4} \int_0^t \|U_k \nabla f(y^k(s)) - U_k g_z(y^k(s))\|^2 ds \right].$$

Setting $t = \lambda_k$, we can express $\mathrm{D}_{\mathrm{KL}}(\mu_t^k \| \nu_t^k)$ as

$$\mathrm{D}_{\mathrm{KL}}(\mu_t^k \| \nu^k) = -\int d\mu_t^k \log \frac{d\nu_t^k}{d\mu_t^k} = \sum_{i=1}^k \mathbb{E}\left[\frac{\beta}{4}\int_0^\lambda \|U_k \nabla f(y^k(s)) - U_k g_z(y^k(s))\|^2 ds\right].$$

since $\int_0^t \left\langle U_k \nabla f(y^k(s)) - U_k g_z(y^k(s)), dW_s \right\rangle$ is a 0-mean Martingale.

Using Assumption 3 and Lemma 9, we obtain

$$\begin{aligned}
\mathrm{D}_{\mathrm{KL}}(\mu_t^k \| \nu_t^k) &= \sum_{i=1}^k \mathbb{E}\left[\frac{\beta}{4}\int_0^\lambda \|U_k \nabla f(y^i(s)) - U_k g_z(y^i(s))\|^2 ds\right] \\
&\leq \sum_{i=1}^k \left[\frac{\beta}{4}\int_0^\lambda M^2 \mathbb{E}\|y^i(s)\|^2 + B^2 ds\right] \\
&\leq \sum_{i=1}^k \left[\frac{\beta}{4}\int_0^\lambda M^2(e^{-ms}\mathbb{E}\|y^i(0)\|^2 + \frac{d_i/\beta + c}{m}(1 - e^{-ms})) + B^2 ds\right].
\end{aligned}$$

where we have applied Lemma 9 in the last line. Integrating, we obtain

$$\mathrm{D}_{\mathrm{KL}}(\mu_t^k \| \nu_t^k) \leq \sum_{i=1}^k \frac{\beta}{4}\mathbb{E}\left[\frac{M^2}{m}(1 - e^{-m\lambda})\mathbb{E}\|y^i(0)\|^2 + \frac{M^2(d_i/\beta + c)}{m^2}(mt + e^{-m\lambda} - 1)) + B^2\lambda\right].$$

Expanding $e^{-m\lambda}$ and leveraging that $m\lambda \geq 1 - e^{-m\lambda} \geq m\lambda - \frac{m^2\lambda^2}{2}$

$$\mathrm{D}_{\mathrm{KL}}(\mu_\lambda^k \| \nu_\lambda^k) \leq \sum_{i=1}^k \frac{\beta M^2 \lambda}{4}\mathbb{E}\|y^i(0)\|^2 + \frac{M^2\lambda^2(d_i + c\beta)}{4} + \frac{\beta B^2 t}{4}.$$

By repeatedly expanding Lemma 9, we obtain

$$\begin{aligned}
\mathrm{D}_{\mathrm{KL}}(\mu_t^k \| \nu_t^k) &\leq \sum_{i=0}^{k-1} \frac{M^2\beta\lambda}{4}\kappa_0 + e^{-m(i-1)\lambda}\frac{M^2\lambda^2(d_i + \beta c)}{8} + \frac{M^2\lambda^2(d_i + \beta c)}{8} + \frac{\beta B^2 \lambda k}{4} \\
&\leq \frac{M^2\beta\lambda k}{4}\kappa_0 + \frac{M^2\lambda^2(d_{\max} + \beta c)k}{4} + \frac{\beta B^2\lambda k}{4} \\
&\triangleq (C_1 + C_2\lambda)\lambda k.
\end{aligned}$$

where we have defined for convenience

$$C_1 \triangleq \frac{M^2\beta\kappa_0}{4} + \frac{\beta B^2}{4}$$

and

$$C_2 \triangleq \frac{M^2(d_{\max} + \beta c)}{4}$$

By Lemma 10, we can bound $W_2^2(\mu_t^k, \nu_t^k)$ as

$$W_2^2(\mu_t^k, \nu_t^k) \leq 4C^2\left[\mathrm{D}_{\mathrm{KL}}(\mu\|\nu)^{1/2} + \left(\frac{\mathrm{D}_{\mathrm{KL}}(\mu\|\nu)}{2}\right)^{1/4}\right]^2.$$

Setting $\alpha = 1$, $d(x) = \|x\|^{1/2}$, and $p = 1/2$, we obtain from Lemma 11

$$4C^2 \leq (12 + 4\kappa_0 + 8(2c + \frac{d_{\max}}{\beta})k\lambda).$$

Note that for any $a \geq 0$, we have $(\sqrt{a} + (\frac{a}{2})^{1/4})^2 \leq 2a + 2\sqrt{a}$, since

$$(\sqrt{a} + (\frac{a}{2})^{1/4})^2 = a + 2^{3/4}a^{3/4} + \frac{a^{1/2}}{2^{1/2}} = a + (2^{1/4}a^{1/4})(2^{1/2}a^{1/2}) + \frac{a^{1/2}}{2^{1/2}}.$$

By Young's inequality, $(2^{1/4}a^{1/4})(2^{1/2}a^{1/2}) \leq \frac{\sqrt{a}}{2^{1/2}} + a$, hence

$$(\sqrt{a} + (\frac{a}{2})^{1/4})^2 = a + (2^{1/4}a^{1/4})(2^{1/2}a^{1/2}) + \frac{a^{1/2}}{2^{1/2}} \leq 2a + \frac{2\sqrt{a}}{\sqrt{2}} \leq 2a + 2\sqrt{a}.$$

plugging in Lemma 11, and assuming $k\lambda \geq 1$, $k > \lambda$ we have

$$W_2^2(\mu_t^k, \nu_t^k) \leq 2C^2 \left[ D_{KL}(\mu\|\nu) + \sqrt{D_{KL}(\mu\|\nu)} \right]$$

$$\leq (12 + 8(\kappa_0 + (2c + d_{max}/\beta))) \left[ (C_1 + C_2\lambda)k\lambda + \sqrt{(C_1 + C_2\lambda)}k\lambda \right] (k\lambda)$$

$$\leq (12 + 8(\kappa_0 + (2c + d_{max}/\beta))) \left[ (C_2 + \sqrt{C_2})\sqrt{\lambda k} + (C_1 + \sqrt{C_1}) \right] (k\lambda)^2$$

$$= C_0^2 \left[ (C_2 + \sqrt{C_2})\sqrt{\lambda k} + (C_1 + \sqrt{C_1}) \right] (k\lambda)^2.$$

We thereby obtaining Lemma 2 with:

$$C_0 \triangleq (12 + 8(\kappa_0 + (2c + d_{max}/\beta))),$$
$$C_1 \triangleq \frac{M^2\beta\kappa_0}{4} + \frac{\beta B^2}{4},$$
$$C_2 \triangleq \frac{M^2(d_{max} + \beta c)}{4}.$$

$\square$

## E.2 Constants in Expected Function Gap Bounds

We start by recalling the Lemma from Polyanskiy & Wu (2016):

**Lemma 12** (Wasserstein Continuity for Quadratic-Growth Potentials). *Let $\mu$, $\pi$ be probability distributions with finite second moments and let $f : \mathbb{R}^d \to \mathbb{R}^+$ be a continuously differentiable function satisfying $\|\nabla f(x)\|^2 \leq c_1\|x\|^2 + c_2$. Then we have*

$$\left| \int f(x)d\mu(x) - \int f(x)d\pi(x) \right| \leq (c_1\sigma + c_2)W_2(\mu, \pi)$$

*where $\sigma = \sqrt{\max[\mathbb{E}_\mu[\|x\|^2], \ \mathbb{E}_\pi[\|x\|^2]]}$.*

Raginsky et al. (2017) bound the constant $\sigma^2 = \max \mathbb{E}_{\mu^k}[x^2], \mathbb{E}_\pi[x^2]$ using an unbiased oracle. As discussed in the main text, DXs have fixed device variation from analog errors, precluding unbiased estimation. However, DX errors take the form of perturbations in the underlying function, i.e. the target function characteristics are intact. For instance, DXs with quadratic potential targets (Aifer et al., 2023; Song et al., 2024) are still optmizing/sampling quadratic functions. Accordingly, Assumptions 4 and 5, that the DX gradient retains both Lipschitz continuity and dissipativity, are reasonable. Assuming $g_\delta$ is $(\mathfrak{m}, \mathfrak{c})$-dissipative, we have from Lemma 3 of Raginsky et al. (2017):

$$\|x(t)\|^2 \leq \kappa_0 + \frac{\mathfrak{c} + d/\beta}{\mathfrak{m}}.$$

Then

$$\sigma^2 = \kappa_0 + \max \left[ \frac{c + d/\beta}{m}, \frac{\mathfrak{c} + d/\beta}{\mathfrak{m}} \right].$$

