# OpenReview forum: "Provable Convergence Bounds for Hybrid Dynamical Sampling and Optimization"
_ICLR.cc/2025/Conference — ICLR 2025 Poster_

### Official Review · Reviewer_kqH5 · 2024-10-26

**Soundness:** 2
**Presentation:** 1
**Contribution:** 2
**Rating:** 3
**Confidence:** 2

**Summary:**

The authors analyze analog accelerators and large-neighborhood local search (LNLS) frameworks. Reducing LNLS to block Langevin Diffusion algorithms, the paper provides convergence guarantees using the tools from the classical sampling theory.

**Strengths:**

Before I start my review, I should acknowledge that the topics of this paper, including Langevin Diffusion (BLD) algorithms, Analog dynamical accelerators, SDEs, LNLS frameworks, are very different from what I do in my research. My main field of interest is mathematical optimization.

I have no doubt that analog computations are an important direction to accelerate the current expensive digital algorithms: the topic is important and relevant. The author's attempt at approaching the issue is unusual (in a good way) and nontrivial.

**Weaknesses:**

The main weakness is that it is challenging to read the paper. From the beginning, the authors introduce many uncommon words and terms that are very unlikely to be easily understood by most researchers from the ICLR community. I think the introduction and the background should be significantly simplified for a broad audience. For instance, the main object of interest is LNLS, but the authors do not try to explain the mathematical foundation and the background of LNLS. Figure 1 is too abstract to understand LNLS.

Other weaknesses and questions:
1. Why do you consider Block Langevin Diffusion? Why can't we optimize w.r.t. all variables?
2. Lines 345-347: I guess there should be $||x - y||^2$ instead of $||x^2 - y^2||$
3. Assumption 5: How does the function inside the integral depends on $t$?
4. Assumption 6: In my experience, this is a very *uncommon* assumption. Also, Assumption 3 is also very uncommon.
5. Theorem 3: This theorem yields the convergence rate $\log \frac{1}{\varepsilon} + \varepsilon,$ which is $\geq 1.$ What If one wants to make the Wasserstein distance less or equal $0.001$?

Unfortunately, reading this paper, I'm not convinced that the reduction to Langevin Diffusion algorithms can not help to improve and explain analog accelerators. At the same time, I do not have expertise in these fields, so I choose low confidence.

**Questions:**

(see weaknesses)

---

> ### Author Response · Authors · 2024-11-20
> **Author Response (See follow-up comment for reference list)**
>
> Thank you for your review, and we appreciate that you saw novelty in our approach. In line with your critique, we have attempted to reduce some of the unnecessarily technical language, and have clarified the LNLS framework in our revised manuscript. We hope that these changes and our responses to your questions assuage your doubts. Due to space constraints, we cannot provide background entirely to our satisfaction. However, we believe there are significant sub-sets of the ICLR community familiar with sampling analysis, analog neural networks, or unconventional computing systems who would find our work useful and insightful, particularly given the uptick in non-von Neumann computing approaches in recent years.
>
> Answers to Questions:
> > Why do you consider Block Langevin Diffusion? Why can't we optimize w.r.t. all variables?
>
> Dynamical accelerators have a finite device capacity. However, real-world problems will often exceed that capacity, requiring hybrid algorithms such as LNLS to partition the problem into DX-amenable subproblem ``blocks'', hence our proposed block Langevin diffusion model. Ideally we would be able to fit the entire problem onto the DX and optimize all variables concurrently. We consider block-partitioned convergence out of necessity, given the widespread use of LNLS by the DX community. Upon re-reads, we realized that our description of LNLS was unnecessarily technical, hence we have simplified the language (see lines 53-54, 227-230).
>
>
> > Lines 345-347: I guess there should be $\Vert x-y\Vert^2$ instead of $\Vert x^2-y^2\Vert$
>
> Yes, thank you for noticing. We've simplified the notation in line with our later use of the Lipschitz constants, and have removed the exponents entirely.
>
>
> > Assumption 5: How does the function inside the integral depends on $t$?
>
> Our original meaning was that the second moment of the iterate gradient was exponentially integrable, so $x$ should have been written $x(t)$. However, we greatly thank you for raising this question, as we realized in revisions that this assumption was superfluous and only muddied the waters. In our revised article we have removed Assumption 5 and simply assume that the gradient oracle is also dissipative, as we discussed in Appendix D.2 in any case.
>
> > Assumption 6: In my experience, this is a very uncommon assumption. Also, Assumption 3 is also very uncommon.
> We apologize if we were unclear in our meaning, however we politely disagree. Both assumptions (or similar) are common in stochastic/non-convex optimization and sampling works. Here we provide a (non-exhaustive) list of examples. Assumption 3 is usually expressed as bounded oracle variance/bias, see Raginsky et al. 2017, Dalalyan and Karagulyan 2019, Chen et al. 2020, Zou and Gu 2021, and Seok and Cho 2023. Several other works consider more restrictive assumptions on the gradient oracle, such as sub-Gaussian tails (Mou et al. 2018, Pensia et al. 2022).
>
> The dissipativity assumption has been used in Raginsky et al. 2017, Xu et. al 2018, Zou et al. 2021, and Farghly and Rebeschini 2021. This assumption requires that the objective function is strongly convex outside of a bounded region, but allows for non-convexity within that region, and is therefore mainly used in works centering on global optimization within non-convex landscapes. It likely does not appear within most mathematical optimization literature, which tends to focus on convex optimization or on convergence to local stationary points within non-convex problems.
>
> > Theorem 3: This theorem yields the convergence rate $\log\frac{1}{\varepsilon} +\frac{1}{\varepsilon}$ which is $\geq 1$ What If one wants to make the Wasserstein distance less or equal $0.001$?
>
> Thank you for pointing out our lack of explanation. Like the traditional unadjusted Langevin algorithm, non-ideal BLD is asymptotically biased: there exists a finite lower bound for the $W_2$ distance to the target measure. In traditional Langevin Monte Carlo, this bias is due to the ``forward-flow'' operator splitting scheme with finite step size (see Wibisono 2018 for an excellent presentation of this topic). In the case of our analysis, the bias is due to analog component variation, since the bias constants are proportional to the non-ideality parameters $M$, $B$.

---

> > ### Author Response · Authors · 2024-11-20
> > **Referenced Works**
> >
> > References:
> >
> > [1] Ji. Seok and C. Cho, “Stochastic Gradient Langevin Dynamics Based on Quantization with Increasing Resolution,” Oct. 04, 2023, arXiv: arXiv:2305.18864. doi: 10.48550/arXiv.2305.18864.
> >
> >
> > [2] D. Zou and Q. Gu, “On the Convergence of Hamiltonian Monte Carlo with Stochastic Gradients,” in Proceedings of the 38th International Conference on Machine Learning, PMLR, Jul. 2021, pp. 13012–13022. Accessed: Nov. 29, 2023. [Online]. Available: https://proceedings.mlr.press/v139/zou21b.html
> >
> >
> > [3] T. Farghly and P. Rebeschini, “Time-independent Generalization Bounds for SGLD in Non-convex Settings,” presented at the Advances in Neural Information Processing Systems, Nov. 2021. Accessed: Nov. 13, 2024. [Online]. Available: https://openreview.net/forum?id=tNT4APQ0Wgj
> >
> > [4] X. Chen, S. S. Du, and X. T. Tong, “On Stationary-Point Hitting Time and Ergodicity of Stochastic Gradient Langevin Dynamics,” Journal of Machine Learning Research, vol. 21, no. 68, pp. 1–41, 2020.
> >
> >
> > [5] A. S. Dalalyan and A. Karagulyan, “User-friendly guarantees for the Langevin Monte Carlo with inaccurate gradient,” Stochastic Processes and their Applications, vol. 129, no. 12, pp. 5278–5311, Dec. 2019, doi: 10.1016/j.spa.2019.02.016.
> >
> >
> > [6] P. Xu, J. Chen, D. Zou, and Q. Gu, “Global Convergence of Langevin Dynamics Based Algorithms for Nonconvex Optimization,” in Advances in Neural Information Processing Systems, Curran Associates, Inc., 2018. Accessed: May 17, 2023. [Online]. Available: https://proceedings.neurips.cc/paper/2018/hash/9c19a2aa1d84e04b0bd4bc888792bd1e-Abstract.html
> >
> > [7] A. Pensia, V. Jog, and P.-L. Loh, “Generalization Error Bounds for Noisy, Iterative Algorithms,” in 2018 IEEE International Symposium on Information Theory (ISIT), Jun. 2018, pp. 546–550. doi: 10.1109/ISIT.2018.8437571.
> >
> > [8] W. Mou, L. Wang, X. Zhai, and K. Zheng, “Generalization Bounds of SGLD for Non-convex Learning: Two Theoretical Viewpoints,” in Proceedings of the 31st  Conference On Learning Theory, PMLR, Jul. 2018, pp. 605–638. Accessed: Nov. 13, 2024. [Online]. Available: https://proceedings.mlr.press/v75/mou18a.html
> >
> > [9] M. Raginsky, A. Rakhlin, and M. Telgarsky, “Non-convex learning via Stochastic Gradient Langevin Dynamics: a nonasymptotic analysis,” in Proceedings of the 2017 Conference on Learning Theory, PMLR, Jun. 2017, pp. 1674–1703. Accessed: Nov. 11, 2023. [Online]. Available: https://proceedings.mlr.press/v65/raginsky17a.html
> >
> > [10] A. Wibisono, “Sampling as optimization in the space of measures: The Langevin dynamics as a composite optimization problem,” in Proceedings of the 31st  Conference On Learning Theory, PMLR, Jul. 2018, pp. 2093–3027. Accessed: Feb. 29, 2024. [Online]. Available: https://proceedings.mlr.press/v75/wibisono18a.html
> >
> > [11] D. Zou, P. Xu, and Q. Gu, “Faster Convergence of Stochastic Gradient Langevin Dynamics for Non-Log-Concave Sampling,” in Proceedings of the Thirty-Seventh Conference on Uncertainty in Artificial Intelligence, PMLR, Dec. 2021, pp. 1152–1162. Accessed: Nov. 14, 2024. [Online]. Available: https://proceedings.mlr.press/v161/zou21a.html

---

> ### Comment · Reviewer_kqH5 · 2024-11-21
>
> Thank you.
>
> 1.
> > In line with your critique, we have attempted to reduce some of the unnecessarily technical language, and have clarified the LNLS framework in our revised manuscript. We hope that these changes and our responses to your questions assuage your doubts.
>
> Can you please highlight the changes in blue (or any other color)? I want to see if the changes make the understanding easier.
>
> 2.
>
> Is there some relation of Assumption 5 (in the revision) to the strong convexity? How are they connected? For a strongly convex function $f,$ we have $\langle \nabla f(x) - \nabla f(x^*), x - x^* \rangle \geq m ||x - x^*||^2,$ where $x^*$ is the minimum of $f.$ For $c = 0,$ why is $x^* = 0$ in Assumption 5?

---

> > ### Author Response · Authors · 2024-11-21
> > **Author Response**
> >
> > > Can you please highlight the changes in blue (or any other color)? I want to see if the changes make the understanding easier.
> >
> >
> > We have uploaded a `latexdiff` version of the manuscript with changes highlighted in blue. We hope that this aids in the review process.
> >
> > > Is there some relation of Assumption 5 (in the revision) to the strong convexity? How are they connected? For a strongly convex function $f$, we have $\langle \nabla f(x)-\nabla f(x^*),x-x^*\rangle\geq \lVert x-x^*\rVert^2$, where $x^*$ is the minimum of $f$. For $c=0$, why is $x^*=0$ in Assumption 5?
> >
> > Thank you for pointing out an omission on our part, which we have rectified in the revised manuscript. We assumed without loss of generality that $\min f(x)=0$ with $x^*=0$, since we can simply add a translation to the function to satisfy this condition (which doesn't affect the first-order algorithm).
> >
> > As for the relation between dissipativity and strong convexity, Assumption 5 is equivalent to saying that the function is non-convex inside of a bounded region and $m$-strongly convex outside of that ball. In this case, $c$ is the maximum deviation from the strong convexity condition inside the ball. See, for example, Ma et al. 2019 ``Sampling can be faster than optimization'' for the an example of this alternative bounded region definition. We use dissipativity rather than bounded non-convexity due to our use of Raginsky et al. 2017's mathematical framework, however the latter definition is more intuitive.

---

### Official Review · Reviewer_YcU8 · 2024-10-29

**Soundness:** 3
**Presentation:** 3
**Contribution:** 3
**Rating:** 8
**Confidence:** 1

**Summary:**

This paper presents the first explicit probabilistic convergence guarantees for hybrid Langevin Noise Likelihood Sampling (LNLS) algorithms in activation sampling and optimization. The authors reduce hybrid LNLS to block sampling using continuous-time Langevin diffusion sub-samplers, analyzing randomized and cyclic block selection rules. They demonstrate that ideal accelerators converge exponentially under a log-Sobolev inequality, while finite device variation introduces bias in the Wasserstein distance. Numerical experiments on a toy Gaussian sampling problem illustrate the effects of device variation and hyperparameters.

**Strengths:**

- The paper is clearly structured, with each theorem building on the previous results to form a coherent narrative.

-  The findings of the paper are supported by clear numerical experiments.

**Weaknesses:**

N/A

**Questions:**

N/A

---

> ### Author Response · Authors · 2024-11-20
> **Author Response**
>
> Thank you for your favorable review, we were heartened to hear that you found our presentation well-structured and coherent, and that our numerical experiments provided further clarity.

---

### Official Review · Reviewer_o8aG · 2024-10-29

**Soundness:** 4
**Presentation:** 3
**Contribution:** 2
**Rating:** 8
**Confidence:** 2

**Summary:**

Analogue accelerators are attracting renewed interest, promising far superior power efficiency and latency compared to digital methods for problems in machine learning, optimization, and sampling.
While the theoretical understanding of analogue accelerators has evolved quickly, significant gaps remain when taking into account a fundamental practical aspect of those devices:
their limited capacity makes it necessary to solve larger problems "piece-by-piece".
That is, the device operates on a subset of the problem at a time while keeping the rest constant, progressively iterating over the entire problem.

The authors find a rich connection between this constraint and the theory of block Langevin diffusion algorithms.
With the connection to well-established theory, the authors adapt existing methods to obtain novel bounds on the performance of a class of hybrid analogue-digital algorithms and non-asymptotic guarantees for their convergence when accounting for non-ideal devices (which are inevitable in practice).

**Strengths:**

1. The paper is well-written.
    The exposition is clear with remarkably few typos, the motivation is put clearly, the authors bring and discuss relevant limitations, and they provide some discussion after presenting design choices, results, and new ideas, in general.
    They also highlight key ideas underlying their proofs.

2. The work is decently contextualized.
    The contribution relies significantly on many existing works, which the authors appear to recognize and discuss fairly.

3. The authors are upfront and honest about the limitations of their work.

4. The general topic is interesting and timely.

5. The reduction to block Langevin diffusion seems like a natural (and, thus, promising) approach to the problem.
    I believe it should motivate several follow-up works.
    The approach also yields significantly softer assumptions compared to similar previous results.

6. The results feature valuable properties for practical applications, such as explicit constants, hyperparameter simplification, and the handling of some device variation.

**Weaknesses:**

1. The care mentioned in strength (1) does not extend to the appendices. E.g., Appendix A would greatly benefit from some discussion about the impact and intuitions behind the choices made for the experiments.

2. The last sentences of the paragraph 051-059 ask for some substantiation, but the authors offer no references to back them up. Some further discussion could also solve this issue.

3. Despite strength (2) and my comprehension of the space constraints, I believe the paper relies too heavily on references to explain the concept.
    I do not see the reliance on previous works as a problem in general, as much of it is a side effect of the strong fruitful connection the authors made with consolidated theory.
    Still, at points such as Section 4, I felt like essential details were left to be found in the references.
    I am sure there is some curse of knowledge at play here, which is understandable, but it would be a good use of the authors' sharp eloquence to make the paper a bit more self-contained.
    I apologize for not substantiating this claim with specific examples, but it is hard to do it when my point is precisely that I did not get a good grasp of what was being presented.

4. The role of analogue-to-digital conversion is not discussed.
    While I am not sure how pertinent this is for this particular work (see question 1), ADC bottlenecks are so common in analogue computing that it should deserve at least a mention.

---
### Minor issues and suggestions

I'll only mention typographical matters because I noticed the authors were particularly zealous with that —I spotted the 2-letter `\emph` on line 269!
They did a great job, overall, and those suggestions aim to help them further improve their skills.

1. Consider numbering only the equations that are referenced in the text (rather than all of them).
2. Most colons preceding equations should be removed.
    More generally, equations are an integral part of the text, reading as sentences.
    This also means that equations should be punctuated as such (this issue affects almost all equations in this work).
    Any maths style guide would serve as reference for this. As an example, Section 13.4 of the AMS Style Guide (I'd provide a link, but this is disallowed for reviewers) mentions both issues.
3. Some `\mathrm`s and `\operatorname`s are missing.
    See, for instance, Assumption 1.
4. By eyeballing, I suspect the authors use `||` (double vertical bars) when they should use `\lVert`, `\rVert`, or `\Vert` which ensure proper spacing.
    For example, compare $||x||$ and $\lVert x \rVert$ (the latter is the correct one) or $\mathrm{D}_{\mathrm{KL}}(\mu || \pi)$ and $\mathrm{D_{KL}}(\mu \mathrel{\Vert} \pi)$ with the latter being coded as `\mathrm{D_{KL}}(\mu \mathrel{\Vert} \pi)`.
5. At 235, having the domain of $i$ specified in the definition of $\overline{B}_i$ would be helpful.
6. At 105, mentioning $\beta$ is premature.
    The sentence also references Equation 22 which is 7 pages ahead!
7. In Assumptions 3 and 4, the domain of $\delta$ (denoted by $\mathbf{D}$) is not defined.

**Questions:**

1. In the applications familiar to me, analogue-to-digital conversion tends to be an crucial bottleneck for hybrid analogue/digital accelerators.
    This affects their accuracy, latency, power efficiency, and, most crucially, die footprint which largely determines their cost.
    ADCs are so expensive in so many ways that many applications sacrifice as much precision as possible to minimize their use.

    In this light, how are those aspects relevant to your work?
    Do the experiments take them into account?
    Do previous works on the topic address them?


2. As the authors say, performing experiments with Gaussian distributions allows for closed-form solutions for the 2-Wasserstein distance. Yet, even though the plots from Figure 2 display a $y$-axis with units, it is hard to reason quantitatively reason in terms of $W_2$. Could you provide some general guidance for that? I mean, is a $W_2$ of 1 large? I understand this can be problem-dependent, but some general guidance would be helpful.

---

> ### Author Response · Authors · 2024-11-20
> **Author Response (Part 1 of 2)**
>
> Thank you for your insightful comments, helpful suggestions, and praise of our work. We are particularly grateful for the typographical advice, and we have fixed all of the minor formatting concerns that you raised. Suffice to say, several authors now have the AMS style guide bookmarked for future reference.
>
> Answers to Questions:
>
> > In the applications familiar to me, analogue-to-digital conversion tends to be an crucial bottleneck for hybrid analogue/digital accelerators. This affects their accuracy, latency, power efficiency, and, most crucially, die footprint which largely determines their cost. ADCs are so expensive in so many ways that many applications sacrifice as much precision as possible to minimize their use.}
> In this light, how are those aspects relevant to your work? Do the experiments take them into account? Do previous works on the topic address them?
> As you correctly note, ADCs incur significant latency, power, and area bottlenecks, leading to the widespread adoption of low-precision output representations ($\leq 8b$). The error introduced by low-precision iterates is certainly relevant to this work. The closest work that we are aware of is a recent pre-print (Seok and Cho 2023), however that work considered intentional gradient quantization rather than optimizing low-precision iterates.
>
> While we could include quantization gradient error under Assumption 3, bounding the asymptotic Wasserstein bias from sampling quantized iterates is less straightforward. Moreover, quantization may also impose lower bounds on the sampling time per block, since the DX state needs to change beyond the detectable precision of the ADC to make forward progress. Given that these are highly non-trivial research problems, we leave consideration to future work. Accordingly, we have added ADC-incurred precision loss to the ``Limitations'' section.
>
>
> > As the authors say, performing experiments with Gaussian distributions allows for closed-form solutions for the 2-Wasserstein distance. Yet, even though the plots from Figure 2 display a $y$-axis with units, it is hard to reason quantitatively reason in terms of $W_2$. Could you provide some general guidance for that? I mean, is a $W_2$ of 1 large? I understand this can be problem-dependent, but some general guidance would be helpful.
>
> We agree that high-dimensional $W_2$ is not the most intuitive metric. Our focus in the numerical experiments was to compare *rates* of convergence rather than focusing on precise values. E.g., the block methods are slower than full LD with the expected dependence on block size and step duration. We have clarified this focus before discussing results in Section 4.
>
>
> References:
>
>
> [1] Ji. Seok and C. Cho, “Stochastic Gradient Langevin Dynamics Based on Quantization with Increasing Resolution,” Oct. 04, 2023, arXiv: arXiv:2305.18864. doi: 10.48550/arXiv.2305.18864.

---

> > ### Author Response · Authors · 2024-11-20
> > **Author Response (Part 2 of 2)**
> >
> > Addressing Concerns:
> >
> > > The care mentioned in strength (1) does not extend to the appendices. E.g., Appendix A would greatly benefit from some discussion about the impact and intuitions behind the choices made for the experiments.
> >
> > We have added more details and discussion regarding our experimental motivations, setup, and analysis in Appendix A. We have also added more details and discussion for our results in the appendices to ease readability.
> >
> > > The last sentences of the paragraph 051-059 ask for some substantiation, but the authors offer no references to back them up. Some further discussion could also solve this issue.
> >
> > We have added references device variation studies from analog neural network literature to substantiate our claims, and have noted both retraining and hyperparameter adjustment as potential costs of accelerator migration.
> >
> > > Despite strength (2) and my comprehension of the space constraints, I believe the paper relies too heavily on references to explain the concept. I do not see the reliance on previous works as a problem in general, as much of it is a side effect of the strong fruitful connection the authors made with consolidated theory. Still, at points such as Section 4, I felt like essential details were left to be found in the references. I am sure there is some curse of knowledge at play here, which is understandable, but it would be a good use of the authors' sharp eloquence to make the paper a bit more self-contained. I apologize for not substantiating this claim with specific examples, but it is hard to do it when my point is precisely that I did not get a good grasp of what was being presented.
> >
> > We thank you for pointing out this particular weakness, as another reviewer also noted that portions of our work lacked accessibility without familiarity of the works being cited. Accordingly, we have tried to reduce our dependence on ``paper pointers'', or at least try to summarize the key points in our work to increase our work's self-containment. Specifically:
> >
> > 1. For your specific example, we provide more explanatory material in section 4 regarding our choice of DX baseline (Paragraph starting line 457). Specifically, we provide more context on *what* we use from the referenced works and what those works proposed, namely an analog electronic accelerator with an associated RC time constant (6.2 ns).
> >
> > 2. In the same spirit, we attempt to streamline other reference-heavy aspects of our presentation, including our presentation of LNLS (Lines 51, 215-229) and our brief review of proposed DXs in Section 2 (first paragraph). We have replaced more technical language which heavily relies on source material familiarity. For example *"In hybrid LNLS, continuous analog phases are interrupted by discrete control logic to synchronize and switch partitions"* is an overly-technical statement, and was replaced by *"In hybrid LNLS, the DX is used to perform alternating sampling/minimization over within-capacity subproblems."* A person unfamiliar with hybrid Ising machine/DX literature would probably need to read our referenced works to easily draw the second meaning out of the first, motivating the change.
> >
> > We hope that these changes address this particular concern.

---

> > > ### Comment · Reviewer_o8aG · 2024-11-26
> > > **I keep my score, but I think Reviewer kqH5 has a point**
> > >
> > > I thank the authors for their reply and effort to improve the paper. I will keep my score.
> > >
> > > Still, I highlight that the discussion around the match between this work and the venue (especially with Reviewer kqH5) seems to be relevant, indeed. Not that the paper does not fit the scope of the venue, but looking at the number of reviews this work got and the widespread low confidence scores, it seems that the intersection of people familiar with both sampling analysis and analogue neural networks is harder to find than the authors anticipated.
> > > This experience should motivate the authors to give their work's presentation a higher priority in the future.
> > > Hadn't the authors improved the manuscript in this aspect, I would have decreased my score.

---

### Official Review · Reviewer_zAh3 · 2024-11-02

**Soundness:** 3
**Presentation:** 3
**Contribution:** 3
**Rating:** 6
**Confidence:** 1

**Summary:**

I am not engaged in research related to this problem, so I am unable to provide an
objective evaluation on this topic. Please disregard my review comments.

**Strengths:**

N/A

**Weaknesses:**

N/A

**Questions:**

1. In this paper, the authors assume that a vector can be decomposed using tensor products or Kronecker products. However, this decomposition does not span the entire Hilbert space, which implies that the conclusions presented in the paper lack generality.

2. All equations lack punctuation and should be corrected.

---

> ### Author Response · Authors · 2024-11-20
> **Author Response**
>
> We thank you for your review, and we have made revisions to address the concerns that you have raised. We have added punctuation to all of our mathematical statements, as other reviewers also raised this issue. To clarify, our work does not assume that the vectors can be decomposed into tensor products. Rather, we simply assume that we are dealing with a product space (namely $\mathbb{R}^d$) which can be decomposed into subspaces. We have replaced the tensor product operator with a Cartesian product operator to clarify this point (line 215).

---

### Official Review · Reviewer_wnfD · 2024-11-04

**Soundness:** 2
**Presentation:** 3
**Contribution:** 3
**Rating:** 8
**Confidence:** 2

**Summary:**

This paper concerns analysis of hybrid large neighborhood local search (LNLS) frameworks, in which the authors provide non-asymptotic convergence guarantees for this framework. In particular, an exponential non-asymptotic bound is obtained for the KL divergence of DXs employing two different strategies (randomized and cyclic block) and a bias bound on the 2-Wasserstein distance is established for finite device variation. Numerical experiments supporting the theoretical results developed.

**Strengths:**

This is an interesting paper and I believe the contributions are novel. The authors provide a good literature review and contextualization of their results with respect to the past literature. Moreover, the authors did a good job in identifying the limitations of their work.

The paper is objective and its contributions are clearly identified.

I did not have time to review all proofs in detail.

**Weaknesses:**

-I believe that a discussion on the performance differences between Random and Cyclic block approaches would be good for clarification (see questions).

**Questions:**

- Can the authors provide further examples of distributions that would satisfy the LSI? How realistic is that assumption in applications?

- Random and Cyclic block approaches seem to produce similar outcomes. What is the motivation to choosing one over another? Is there any intuition on which one should I choose based upon my application?

- In Figure 2 (e), why doesn't the curve associated with \delta=0 match the ideal curve?

---

> ### Author Response · Authors · 2024-11-20
> **Author Response**
>
> Thank you for your kind review. We were particularly encouraged to hear that you found our work interesting and novel, and that you thought our literature review and discussion praiseworthy. We have attempted to address your questions in our revised draft as well as directly addressing them below:
>
>
> > Can the authors provide further examples of distributions that would satisfy the LSI? How realistic is that assumption in applications?
>
> Distributions of practical interest satisfying an LSI include high-temperature spin systems (of particular interest within combinatorial optimization), globally strongly log-concave measures with bounded regions of non-log concavity (such as weight-decay regularized machine learning), and log-concave measures which are not strongly log-concave (such as heavy tailed exponential distributions). We have also added these examples to Sec. 3 after our introduction of the LSI.
>
> As we note in our "Limitations" section, assuming an LSI is still relatively restrictive. However, we conjecture that our results can be generalized to weaker functional inequalities, potentially using the methods developed by Chewi et al. 2022.
>
> >Random and Cyclic block approaches seem to produce similar outcomes. What is the motivation to choosing one over another? Is there any intuition on which one should I choose based upon my application?
>
> There is no direct convergence reason to favor cyclic over randomized approaches for analog LNLS. However, we believe that cyclic orderings are preferable. Implementations are generally much simpler and are much more straightforward to optimize in practice since the algorithm is more predictable. For instance, we can optimize memory layouts to ensure that contiguous partitions are stored together, reducing overall memory access latency and opening opportunities to exploit the memory hierarchy. Moreover, multi-chip DX proposals such as ``batch mode'' from Sharma et al. 2022 rely on cyclic orderings to implement an efficient hardware pipeline. We have also added notes to this effect in Sec. 3 (lines 286-288) to further motivate our analysis of block orderings.
>
> >In Figure 2 (e), why doesn't the curve associated with $\delta=0$ match the ideal curve?
>
> Thank you for noticing this lack of clear notation. Here ``ideal'' was meant to communicate that the *full* Langevin diffusion had no noise. The $\delta=0.0$ curve represents the *block* Langevin diffusion without noise, which is why it converges more slowly. We have clarified the text in Sec. 4 to make this more explicit.
>
> References:
>
>
> [1] S. Chewi, M. A. Erdogdu, M. Li, R. Shen, and S. Zhang, “Analysis of Langevin Monte Carlo from Poincare to Log-Sobolev,” in Proceedings of Thirty Fifth Conference on Learning Theory, PMLR, Jun. 2022, pp. 1–2. Accessed: Nov. 13, 2024. [Online]. Available: https://proceedings.mlr.press/v178/chewi22a.html
>
> [2] A. Sharma, R. Afoakwa, Z. Ignjatovic, and M. Huang, “Increasing ising machine capacity with multi-chip architectures,” in Proceedings of the 49th Annual International Symposium on Computer Architecture, in ISCA ’22. New York, NY, USA: Association for Computing Machinery, Jun. 2022, pp. 508–521. doi: 10.1145/3470496.3527414.

---

> > ### Comment · Reviewer_wnfD · 2024-11-26
> >
> > I thank the authors for their careful response. I retain my current score.

---

### Author Response · Authors · 2024-11-20
**Revision Uploaded**

Dear Reviewers and AC,

We have uploaded a revised manuscript in an attempt to address reviewer concerns and clarify our presentation. New text is marked in blue to assist with the reviewing process.

The primary content changes are:
- Simplification of some overly technical/reference-heavy portions of the Introduction, Background, and Numerical Experiment sections (in line with comments from **Reviewers o8aG and kqh5**)
- A simplification of our assumptions for Theorem 3. Specifically, we removed the exponential integrability assumption as we realized it was unecessary, and could be replaced by a simpler dissipativity assumption (stemming from a comment by **Reviewer kqh5**).
- Added examples of LSI distributions with associated references to the background section (**Reviewer wnfD**)
- Added further discussion of our experimental methods to Appendix A (**Reviewer o8aG**)

Moreover, we have made a myriad of small typographical/presentation fixes, including properly punctuating our equations, fixing typos, and clarifying figures.

We hope that these changes address the Reviewers' concerns, and we look forward to any further feedback.

---

### Meta-Review · Area_Chair_bdEM · 2024-12-18

**Metareview:**

Summary:
This paper presents a comprehensive analysis of hybrid large neighborhood local search (LNLS) frameworks in the context of analog accelerators, providing non-asymptotic convergence guarantees. The authors reduce LNLS to block sampling using continuous-time Langevin diffusion sub-samplers and analyze both randomized and cyclic block selection approaches. The work establishes exponential convergence under log-Sobolev inequality conditions for ideal accelerators while quantifying bias from finite device variation in the Wasserstein distance.

Main strengths:

- Novel theoretical analysis bridging LNLS with block Langevin diffusion algorithms
- Clear presentation of convergence guarantees with explicit constants
- Thorough experimental validation of theoretical claims
- Practical relevance for analog accelerator implementations
- Careful consideration of real-world constraints like device capacity limitations

Main weaknesses:

- Some sections rely heavily on references without sufficient self-contained explanation
- Technical presentation may be challenging for readers unfamiliar with both sampling theory and analog computing
- Limited discussion of ADC (analog-to-digital conversion) implications
- Appendices could benefit from more detailed experimental discussion

**Additional Comments On Reviewer Discussion:**

Outcomes from author-reviewer discussion:
The authors have been responsive to reviewer feedback and made several improvements:

- Simplified technical language and provided more intuitive explanations of LNLS
- Added clarification regarding practical implications and limitations
- Enhanced discussion of experimental setup and results
- Addressed mathematical notation and formatting issues
- Added references to substantiate claims about device variation

Reviewer agreement/disagreement:
Reviewers generally agreed on the technical merit but differed on accessibility:

- Some found the work well-structured and clearly presented
- Others felt it was too technical for broad accessibility
- There was consensus that the work makes valuable theoretical contributions
- Concerns about venue fit given the specialized topic intersection

Suggestions to improve:

- Further simplify technical presentation for broader accessibility
- Add more intuitive examples and explanations
- Expand discussion of practical implementation considerations

Based on the reviews and author responses, this appears to be a technically strong paper making novel theoretical contributions, though some accessibility concerns remain. The revisions have improved clarity while maintaining technical rigor.

---

### Decision · Program_Chairs · 2025-01-22

Accept (Poster)